# How Language Models Process Negation

Zhejian Zhou [1]   Tianyi Zhou [2]   Robin Jia [2†]   Jonathan May [1†]

## Abstract

We study how Large Language Models (LLMs) process negation mechanistically. First, we establish that even though open-weight models often provide wrong answers to questions involving negation, they do possess internal components that process negation correctly. Their poor accuracy is due to late-layer attention behavior that promotes simple shortcuts; ablating those attention modules greatly improves accuracy on negation-related questions. Second, we uncover how models process negation. We consider two hypotheses: models could use attention heads that attend to the phrase being negated and suppress related concepts, or they could directly construct a representation of the entire negative phrase (e.g., representing "not gas" as a vector that promotes liquids and solids). We apply a range of observational and causal interpretability techniques on Mistral-7B and Llama-3.1-8B to show that models implement both mechanisms, with the "constructive" mechanism being more prominent. Combined, our work deepens the understanding of LLMs' internals, highlighting construction-dominant computations and the coexistence of competing mechanisms within LLMs. Our code is available at https://github.com/Ja1Zhou/LM_Negation.

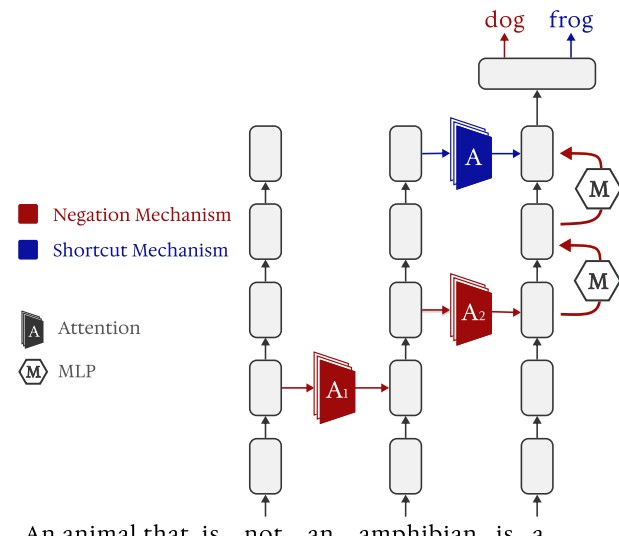

*Figure 1.* Illustration of competing mechanisms for negation. In the negation mechanism, attention module $A_1$ moves the representation of the negation token ("not") to the position of the concept being negated (e.g., amphibian). Subsequently, $A_2$, together with downstream MLPs, constructs and promotes a new negated representation (e.g., mammal). In contrast, the shortcut mechanism bypasses explicit negation reasoning and directly promotes the concept's correlations (e.g., frog).

## 1. Introduction

Negation is ubiquitous in natural language. We aim to elucidate how LLMs compute negation mechanistically. Many prior Mechanistic Interpretability (MI) works focus on settings where the model's final prediction can be explained

by the additive aggregation of factual evidence (Chughtai et al., 2024; Geva et al., 2023; Meng et al., 2022). For example, in the prompt "The Colosseum is in the country of __," "Colosseum" and "country" *independently* contribute to the output "Italy." Negation, however, does **not** fit into this additive paradigm naturally. For the prompt "An animal that is not an amphibian is a __," the model cannot simply combine the tokens promoted by "not" and the tokens promoted by "amphibian." Since "not" can be universally applied to any concept, it does not provide new information on its own that helps determine the output. Negation entails a certain degree of composition, which is scrutinized less in existing literature.

How might LLMs process negation? Prior work suggests two competing hypotheses: **Suppression** or **Construction**. We explain the hypotheses with prompts of the form "X that is not Y is __." An example of such prompt is shown in Figure 1.

†Equal advising. [1]Information Sciences Institute, University of Southern California [2]Thomas Lord Department of Computer Science, University of Southern California. Correspondence to: Zhejian Zhou <zhejianz@usc.edu>, Robin Jia <robinjia@usc.edu>, Jonathan May <jonmay@isi.edu>.

*Proceedings of the 43rd International Conference on Machine Learning*, Seoul, South Korea. PMLR 306, 2026. Copyright 2026 by the author(s).

**Hypothesis 1** (Suppression). *The model promotes the set of tokens that relate to X, and then suppresses the subset of tokens that have property Y.*

**Hypothesis 2** (Construction). *First the model constructs a representation for "not Y." Then, the computed representation $\bar{Y}$ for "not Y" and X triggers latent directions that promote correct answers to the prompt.*

The critical difference between the hypotheses lies in whether the model explicitly constructs a *negated representation* for "not Y." Previous MI works favor Hypothesis 1. Yan & Jia (2025), Wang et al. (2023) and McDougall et al. (2024) discover *negative mover heads* that suppress the tokens that they attend to. On the other hand, the neuroscience literature (Hasson & Glucksberg, 2006; Papeo et al., 2016; Zuanazzi et al., 2024) aligns with Hypothesis 2 and argues that a negated representation is explicitly constructed. From the mechanistic side, Geva et al. (2021) suggest that models favor *promotion* over *suppression*.

In this paper, we first show that current open-weight LLMs (Grattafiori et al., 2024; Yang et al., 2024; 2025; Riviere et al., 2024; Jiang et al., 2023) are **capable** of understanding negation. Inspired by Wang et al. (2023), we define the *sensitivity* metric based on logit differences. Models are sensitive to negation, demonstrating that they must have some mechanism for processing negation (§4.1). Nonetheless, models often provide wrong answers to negative prompts. We show that certain attention heads in late layers are at fault by promoting positive answers on negative prompts. We term these heads shortcut attention heads (Hermann et al., 2024), as they pick up spurious features (e.g., co-occurrence). Ablating late-layer attention modules with our proposed *Attention Sinking* method greatly improves accuracy on negative prompts (§4.2). We further trace the emergence of shortcut attention heads to pre-training (§4.3).

Next, we identify separate model mechanisms that correctly process negation. We find that models use both construction and suppression mechanisms, though *construction* plays a more central role. Through extensive experiments with Llama-3.1-8B[1] and Mistral-7B-v0.1,[2] we uncover how models process "not Y". First, attention moves the representation of "not" to the position of "Y" (§5.1). Next, mid-layer attention modules (§5.2) move a constructed *negated representation* $\bar{Y}$, i.e., a compositional representation of "not Y," to the output position (§5.3). We establish the causal importance of these attention modules via Attention Sinking and path patching, then use LogitLens (nostalgebraist, 2020) to interpret their outputs. For more than 80% of examples in our dataset, we find evidence that these outputs promote concepts related to "not Y". Simultaneously, these same

attention modules suppress the representation of "Y", but to a lesser extent (§5.4). Finally, the constructed representation promotes the correct answer (§5.5): using Sparse AutoEncoders (SAEs; Bricken et al., 2023), we identify MLP output latents that amplify the negated representation. Overall, we reveal that *construction* and *suppression* work collaboratively to compute negation; the model computes a *negated representation* in its mid-layer attention outputs (e.g. interpreting "not gas" as "solid"), while also suppressing the negated concept.

Put together, our contributions are:

- We show that LLMs' failures on negation queries arise not from the absence of negation ability, but because their negation processing mechanisms are overshadowed by other mechanisms at later layers.

- We identify shortcut mechanisms that work against processing negation and introduce Attention Sinking as a simple intervention that mitigates these shortcut attention behaviors.

- We provide a mechanistic account of negation in LLMs, systematically identifying how construction of negated concepts and suppression of the original concepts jointly lead to correct predictions.

## 2. Related Work

**Negation Benchmarking** The study of how well Language Models (LMs) process negation dates back to the era of BERT and RoBERTa (Devlin et al., 2019; Liu et al., 2019). Kassner & Schütze (2020) argue that unsupervised pre-training does not learn negation sufficiently. Gubelmann & Handschuh (2022) and Kletz et al. (2023) refine this claim by showing that some masked encoders are sensitive to negation when providing sufficient context. The seemingly unresolved debate hints to the co-existence of competing mechanisms underlying negation processing. Negation has also been studied in the context of natural language inference (Bowman et al., 2015; Williams et al., 2018). For example, Poliak et al. (2018) and Gururangan et al. (2018) find dataset artifacts related to negation indicators.

Previous benchmarking works focus on evaluating LMs as blackboxes. In this paper, we provide mechanistic answers to both how models process negation and why models sometimes behave as if insensitive to negation.

**Mechanistic Interpretability** Mechanistic Interpretability (Olah et al., 2018) aims to provide finer-grained conclusions about circuit functionalities, often at the level of specific attention heads or Multi-Layer Perceptrons (MLPs). MI techniques have been developed to study the internal representations of LLMs. *Patching* (Meng et al., 2022; Wang et al., 2023) and *LogitLens* (nostalgebraist, 2020) are

---

[1] https://huggingface.co/meta-llama/Llama-3.1-8B
[2] https://huggingface.co/mistralai/Mistral-7B-v0.1

two representatives. *Patching* (*Causal Tracing*) of various forms identifies causally important components by ablating or restoring their activations during forward passes. One exemplary technique is *Attention Knockout* (Geva et al., 2023), which masks out certain tokens in attention to establish their causal effect on the model's prediction. *LogitLens* attempts to interpret internal representations of LLMs by directly projecting them onto the vocabulary.

Recent MI work finds *function vectors* that trigger LLMs to perform certain tasks (Todd et al., 2024), yet do not dive deep into how the function is implemented. In our case of negation, previous work finds a vector that encodes "not," but does not elaborate how "not" composes with "Y" to produce the final answer. Elhelo & Geva (2025) discover attention heads that encode the mapping of antonym pairs or suppress attended tokens. This hints at the existence of both the *construction* and *suppression* mechanisms. We locate causally important attention heads in middle layers, consistent with Skean et al. (2025) who find middle layers essential for transforming representations. On the shortcut mechanism side, Mann et al. (2025) observe that "not X" increases the accessibility of "X" paradoxically. The ineffectiveness of late layers is also observed in Gromov et al. (2025) and Halawi et al. (2024). Building upon existing literature, our study provides finer-grained examinations of negation mechanisms.

## 3. Setup and Background

This section introduces the necessary background for our analysis. We describe (1) the dataset construction, (2) notation conventions used throughout the paper, and (3) the MI methods that are employed in our mechanistic analysis.

### 3.1. Datasets and Notation

**Datasets** We study a controlled family of prompts of the form "X that is not Y is Z." The dataset is defined as

$$\mathcal{D} = \{(P_+^{(n)}, P_-^{(n)}, y_+^{(n)}, y_-^{(n)})\}_{n=1}^N.$$

Here, $P_+$ denotes a *positive* prompt, and $P_-$ denotes its *negated* counterpart, differing only by the insertion of a negation indicator (e.g., not, no, cannot).

The symbols $y_+, y_- \in \mathcal{V}$ denote *single-token* candidate answers drawn from the model vocabulary $\mathcal{V}$, where $y_+$ is the correct answer for $P_+$ and $y_-$ is the correct answer for $P_-$. An example data entry is shown in Table 6.

We format 162 unique questions using 4 prompt templates, generating 648 data entries in total. To provide a better sense of our dataset, we group the subjects of our prompts into categories and provide examples for each category in Table 5. Details of dataset curation are provided in Appendix A.2.

**Residual Stream** We work with Transformer-based models (Vaswani et al., 2017). For internal model activations, we define the following. Let $\mathcal{AO}_i$ denote the attention output vector at layer $i$, $\mathcal{MO}_i$ denote the MLP output vector at layer $i$, and $\mathcal{AP}_i$ denote the attention pattern (i.e., the softmax-normalized attention weights) at layer $i$.

We use superscripts to denote the forward-pass condition. For example, $\mathcal{AO}_i^+$ denotes the attention output at layer $i$ from the forward pass on the positive prompt $P_+$.

Under the standard linearized view of Transformer residual streams, the final hidden state $h_{L+1}$ of a model with $L$ layers can be decomposed as a sum of the embedding and per-layer module outputs:

$$h_{L+1} = E + \sum_{i=1}^{L}(\mathcal{AO}_i + \mathcal{MO}_i). \tag{1}$$

**Logits and Logit Differences** Let $\ell_P(t)$ denote the logit assigned by the model to token $t \in \mathcal{V}$ at the *last token position* of prompt $P$ (i.e., the next-token prediction position). Throughout this paper, all logit-based quantities are evaluated at this position[3].

We define the *logit difference* between two candidate answers $a, b \in \mathcal{V}$ under prompt $P$ as

$$\Delta(P; a, b) := \ell_P(a) - \ell_P(b). \tag{2}$$

### 3.2. Mechanistic Interpretability Preliminaries

**LogitLens** LogitLens is extensively used for MI purposes. For LLMs, the final logits over the vocabulary are produced by passing $h_{L+1}$ through a final layer normalization $\mathcal{LN}_{L+1}$ (typically RMSNorm (Zhang & Sennrich, 2019)), followed by the unembedding matrix $W_U$. If the layer norm scale $\sigma$ is fixed, $\mathcal{LN}_{L+1}(h_{L+1})W_U$ is a linear operation on $h_{L+1}$. Equation 1 suggests that the final logits distribution is an additive ensemble of each component's contribution.

Following the linear view of LLMs, LogitLens projects some feature (e.g. $\mathcal{AO}_{14}$) onto the vocabulary directly. This generates a logits distribution over the vocabulary to help interpret the representation.

**Sparse-AutoEncoder (SAE)** SAEs are trained to reconstruct some representation $x$ using a sparse sum of latents:

$$x \approx \sum_{i=1}^{D} \alpha_i(x) f_i,$$

---

[3]One edge case is that the positive and negative answers are tokenized into multiple tokens. In this case, we evaluate at the first diverging token position between them.

*Table 1.* Attention mass placed on the Attention Sink token set (the first token and the current token) by the last-token of the prompt. For each model, the result is averaged over all layers, heads, and prompts in our dataset. All values are reported as percentages (%) with three significant figures.

| Model | 1st + current (%) |
|---|---|
| Llama-3.1 | 79.5 |
| Qwen2.5 | 68.0 |
| Qwen3 | 70.3 |
| Gemma-2 | 67.9 |
| Mistral-v0.1 | 78.0 |
| OLMo-2 | 63.9 |

*Table 2.* Model performances on our curated dataset.

| Model | Neg Acc | Pos Acc | Sensitivity |
|---|---|---|---|
| Llama-3.1 | 50.5 | 95.2 | 97.4 |
| Qwen2.5 | 57.6 | 93.5 | 96.0 |
| Qwen3 | 55.7 | 91.8 | 95.2 |
| Gemma-2 | 49.7 | 96.5 | 97.5 |
| Mistral-v0.1 | 45.2 | 96.3 | 95.1 |
| OLMo-2 | 54.0 | 96.3 | 97.8 |

where $f$ is the set of learned latents with size $D$ and $\alpha$ is the corresponding coefficients to reconstruct $x$ (Cunningham et al., 2023). For a given $x$, $\alpha(x)$ is a sparse vector.

SAEs can help with interpreting MLPs by cutting down the number of latents to study. Modern LLMs typically have $> 10k$ intermediate size for MLPs. SAEs can cut active latents down to $< 100$.

**Attention Sink**   In this work, we propose and extensively apply the following method to ablate attention modules. Our method is inspired by the work on *attention sinks* (Xiao et al., 2024), which suggests that attending to the first token in the sequence is the "default" behavior of attention heads when they do not perform specific functions. Our *Attention Sink* ablation takes this inspiration. When we **sink** an attention head, we impose a restriction such that the current token can only attend to itself and the first token (the first token is the begin_of_sentence token with no meaningful information). By doing so, we effectively nullify an attention module: it can no longer move information from other positions. However, causal relations such as value matrices or MLPs operating on the current token, are preserved. For a discussion on Attention Sink versus Attention Knockout, see Appendix C.

We further motivate our method with statistics on our dataset, showing that a large amount of attention weight is given to these two tokens, suggesting that our method may not disrupt the model so severely that it collapses. For all prompts in our dataset, we take the last token of the prompt as the *query token* and compute the attention scores assigned to all previous tokens after softmax. The attention scores are normalized between 0 and 1. We sum the scores given to the first and current token (which is the last token of the prompt) and average over all layers, attention heads and prompts. As reported in Table 1, the last-token attention pattern places between 63.9% and 79.5% of its mass on the union of the first token and the current token across all models.

## 4. Shortcut Attention Heads in LLMs

Before diving into our investigation, we first determine whether open-weight LLMs can understand negation in our dataset. Our results are twofold. First, models often output incorrect answers on negative prompts. We observe output failures such as "An animal that cannot fly is a bird." On the other hand, model logits are *sensitive* to negation: models typically prefer $y_+$ to $y_-$ less strongly when given the negative prompt $P_-$ compared with the positive prompt $P_+$. This suggests that some internal mechanism of the model correctly processes negation, but it is overshadowed by another mechanism that promotes $y_+$.

We show that shortcut mechanisms exist for negation and that they are responsible for the observed failures. Besides the fact that models often output wrong answers, there is another piece of evidence: we can largely recover expected behavior (models assigning higher logits to $y_-$ than $y_+$ on $P_-$) by ablating some attention modules. Further, we show that the bias introduced by shortcut attention modules can be traced back to pre-training.

### 4.1. Models Exhibit Internal Sensitivity to Negation

On our curated dataset, we show that LLMs appear to struggle with negation based on *accuracy* metrics. In addition, we define a *sensitivity* metric which reveals that models have learned negation mechanisms.

**Accuracy**   On positive prompts $P_+$, we expect the correct answer $y_+$ to receive a higher logit than the incorrect alternative $y_-$. Conversely, on negated prompts $P_-$, we expect $y_-$ to receive a higher logit than $y_+$. Following the convention established in Section 3, all logits are read out at the *last token position* of the prompt – i.e., the position at which the model produces its next-token prediction. This is the canonical evaluation site for autoregressive LLMs, and it is also the position at which our mechanistic analyses (path patching, Attention Sink ablation, LogitLens) intervene, so accuracy and our circuit-level findings are measured on the same residual stream location.

We measure the *accuracy* of the model as the fraction of prompts for which the logit assigned to the correct answer

*Table 3*. Negative accuracy (%) comparison across models. Applying **Attention Sink** or **LogitLens** to ablate shortcut modules improves negative accuracy. We report the max accuracy achieved across all layers for Attention Sink and LogitLens.

| Model | Full Model | Attn. Sink | LogitLens |
|---|---|---|---|
| Llama-3.1 | 50.5 | **67.8** | 53.6 |
| Qwen2.5 | 57.6 | **65.4** | 59.4 |
| Qwen3 | 55.7 | **64.2** | 59.6 |
| Gemma-2 | 49.7 | **66.1** | 59.7 |
| Mistral-v0.1 | 45.2 | **65.9** | 61.6 |
| OLMo-2 | 54.0 | **68.7** | 61.6 |

exceeds that of the incorrect alternative *at the last token position*. Accordingly, we define **positive accuracy** as this fraction computed over all $P_+$, and **negative accuracy** as the corresponding fraction computed over all $P_-$.

$$\text{Acc}_+ := \frac{1}{|\mathcal{D}|} \sum_{n=1}^{|\mathcal{D}|} \mathbb{I}\left[ \Delta\left( P_+^{(n)}; y_+^{(n)}, y_-^{(n)} \right) > 0 \right]$$

$$\text{Acc}_- := \frac{1}{|\mathcal{D}|} \sum_{n=1}^{|\mathcal{D}|} \mathbb{I}\left[ \Delta\left( P_-^{(n)}; y_-^{(n)}, y_+^{(n)} \right) > 0 \right],$$

where $\mathbb{I}$ denotes the indicator function.

**Sensitivity** Let $\Delta$ denote the logit difference as defined in Eq. (2). *Sensitivity* is defined as:

$$\Pr_{(P_+, P_-, y_+, y_-) \sim \mathcal{D}} [\Delta(P_-; y_-, y_+) > \Delta(P_+; y_-, y_+)].$$

Logits difference directly translates to the probability ratio between two answers. *Sensitivity* measures if over the whole dataset, the probability ratio of $y_-$ to $y_+$ changes in a consistent direction when switching from $P_+$ to $P_-$.

**Results** As shown in Table 2, most models have near-perfect positive accuracies on $P_+$. However, all models are significantly worse on $P_-$. Yet, the *sensitivity* metric consistently reveals that models are *responding* to the presence of negation. We therefore argue that models are reacting to negation internally, but the changes do not manifest themselves at output layers. We further test that *sensitivity* is not an artifact of randomness in Appendix D.1.

### 4.2. Identifying and Mitigating Shortcut Mechanisms

Next, we explain why models have poor negative accuracy despite being sensitive to negation. We find that at the final token position, later layers counterproductively promote positive answer logits on negative prompts. We suspect that late-layer attention modules exhibit shortcut behavior in those layers of the model.

We first show mechanistically that the attention modules are behind the problem. The **Attention Sink** method introduced

in Section 3.2 is employed to both identify and mitigate the shortcut behavior. As explained, sinking attention heads ablates their functionalities. If switching off certain heads recovers expected behaviors on $P_-$, then we conclude that those attention modules play a causally significant role in shortcut behavior. More specifically, we apply *Cumulative Attention Sink* to accomplish our goal. Given some target layer $i$, we sink all attention modules starting from $i$ until the final layer $L$. We provide the motivation for using *Cumulative Attention Sink* in Appendix B.

We also apply LogitLens to show a similar trend. LogitLens directly projects internal representations onto the unembedding matrix to *skip later layers* (which can be viewed as zero-ablating the outputs from later layers). Thus, it also prevents further transformations applied by those layers.

**Results** As presented in Table 3, our Attention Sink ablation consistently improves negative accuracies across models. A similar trend can also be observed with LogitLens, but the improvements are more pronounced for Attention Sink. With our method, we can achieve $17\%$ absolute improvement for Llama-3.1-8B and $46\%$ relative improvement for Mistral-7B-v0.1. In this way, we provide evidence for the existence of a shortcut mechanism in these models. We also validate the effectiveness of our plug-and-play remedy without any additional tuning or collecting additional statistics. For all models, we record the layer at which applying *Cumulative Attention Sink* achieves the best performance. The best layer is consistently $> 0.5L$, suggesting that shortcut modules reside in middle-to-late layers. We provide the best layers to apply Attention Sink and LogitLens for all models in Appendix D.7.

**Sweeping the Sink Layer** To further characterize the role of late-layer attention modules, we sweep the layer at which to apply *Cumulative Attention Sink*. We plot both negative and positive accuracies as a function of the swept layer in Figure 10. Two consistent patterns emerge. First, the attention modules *after* the layer that achieves the best negative accuracy contribute only to positive accuracy. Second, by the best negative accuracy layer, positive accuracy has already nearly saturated to its vanilla (no-sink) value. Together, these observations indicate that the shortcut behavior is concentrated in middle-to-late attention modules and is largely disentangled from the modules that drive positive-prompt accuracy.

### 4.3. Tracing Shortcut Mechanisms

Negative accuracies in Table 2 are $\sim 50\%$. There could be two interpretations: 1) The models are producing close to random accuracies on negative prompts. 2) The models are systematically biasing towards positive answers on negative prompts. We hypothesize that the latter is the case given

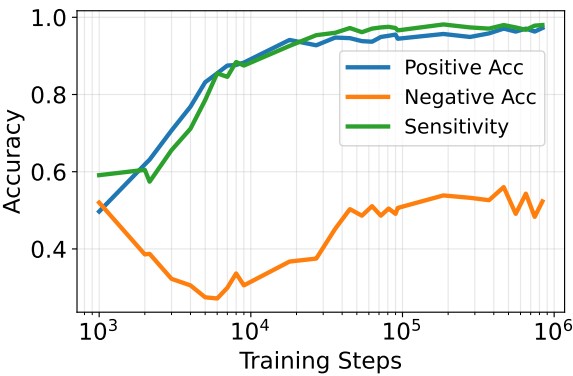

*Figure 2.* OLMo2 Positive and Negative Accuracies at various pre-training checkpoints. We observe that negative accuracy first plummets at early training steps, then rises again and stabilizes.

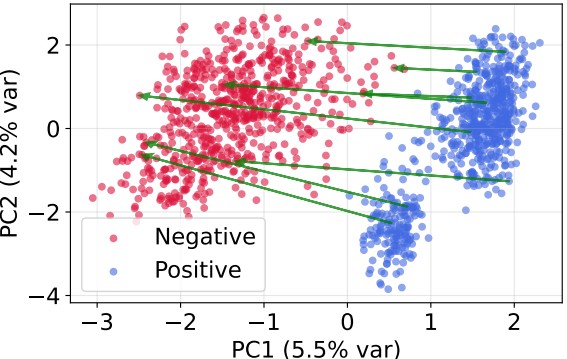

*Figure 3.* Visualization of PCA subspace. The residual stream hidden states are taken from `Llama-3.1-8B` at layer 11 after the attention module (11 mid). The hidden states of $P_+$ and $P_-$ are colored as blue and red. Arrows indicate the direction from one hidden state of $P_+$ to the corresponding hidden state of $P_-$. It can be seen that positive and negative hidden states are approximately linearly separable by one direction.

our discovery of shortcut mechanisms, and that we can find evidence of the emergence of shortcut mechanisms during pre-training. To verify this hypothesis, we test with various checkpoints of OLMo2.

**Accuracies across Train Steps**  We follow our definitions of positive and negative accuracies in Section 4.1 and plot how they evolve across training steps in Figure 2. One revealing observation is that the negative accuracy first plummets at early training steps, then rises again and stabilizes. This suggests that the shortcut attention modules could have formed at early training checkpoints that systematically bias towards outputting positive answers on negative prompts. However, the model is **sensitive** to negation from an early stage following our definition in Section 4.1.

## 5. Mechanisms for Negation

In Section 4.1, we find evidence that negation mechanisms are present and functionally active in LLMs. Next, we analyze how negation is computed by the model's internal modules. We show that LLMs implement *both* suppression and construction mechanisms, with construction playing a more central role: (i) Attention moves the representation of "not" to the position of "Y" in early and middle layers (§5.1). (ii) A negated representation $\bar{Y}$ is **constructed** and moved to the last token position of the input sequence (§5.2, §5.3). (iii) Simultaneously, the representation of "Y" is **suppressed** as it is moved to the last token position (§5.4). (iv) Late-layer MLPs promote the correct answer corresponding to the constructed representation $\bar{Y}$ (§5.5).

### 5.1. Move "Not" to "Y" in Early Layers

The first step of processing the phrase "not Y" is that attention heads move information about "not" to the last token

position of "Y" ("Y" is potentially a multi-token phrase). We hypothesize that this negation signal is manifest in the residual stream such that the positive hidden states $h^+$ and negative hidden states $h^-$ are linearly separable. To verify this hypothesis, we first apply Principal Component Analysis (PCA) following Rimsky et al. (2024) and Marks & Tegmark (2024). PCA helps visualize the data and also serves as feature selection. Then, we perform Linear Discriminant Analysis (LDA) (Fisher, 1936).

**Experiment Pipeline**  The experimental pipeline proceeds as follows. First, for each prompt pair $(P_+, P_-)$, we collect the hidden states at all layers, denoted by $h^+$ and $h^-$, at the last token of the potentially multi-token phrase $Y$. Then, we perform 10-fold cross-validation and divide the dataset into train-test splits. On the training set, we apply PCA to each layer $i$ to reduce the hidden states $h_i^+$ and $h_i^-$ to two dimensions. As illustrated in Figure 3, PCA begins to reveal a consistent "not" direction separating the hidden states of "Y" and "not Y" in the early layers (more illustrations are provided in Figure 11). In the PCA subspace, we fit an LDA model using labels indicating whether a prompt is positive or negative. The LDA model computes a direction that best separates the two classes. We take the direction with the highest training accuracy over all layers, which likely represents "not."

After identifying this direction, we project $h_i^+$ and $h_i^-$ from all layers $i$ onto this direction. For each layer $i$, we compute an LDA model $\mathcal{LDA}_i$ on the training set. Finally, we evaluate $\mathcal{LDA}_i$ at layer $i$ on the test set. We plot the cross-validated accuracy as a function of layer index. A higher accuracy indicates that "not" can be more reliably decoded.

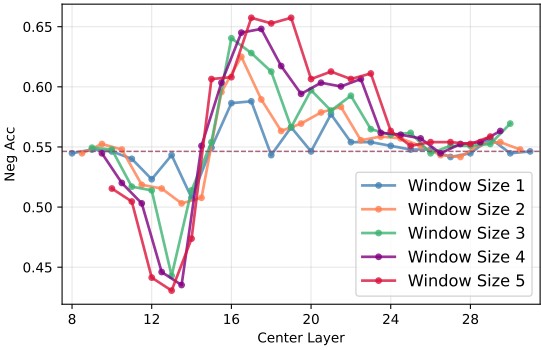

*Figure 4.* Attention Sink results performed on `Llama-3.1-8B`. The x-axis is the center of the window. The y-axis is the negation accuracy over the whole dataset. The dotted line is the negation accuracy of the vanilla model. 1) Attention modules around layer 14 are causally important. 2) Shortcut attention heads around layer 17 are identified. 3) Later layers play little role in negation.

**Results**   As shown in Figure 8, we can decode "not" from the position of "Y" with increasing accuracy. By layer 4, we already achieve close to perfect accuracy. This suggests that early attention layers move "not" to the position of "Y," laying a foundation for further composition.

**Discussion**   Based on PCA results, it is plausible that the residual stream state of "not Y" is an additive combination of the representation of "not" and the representation of $Y$. This view conforms to part of the Linear Representation Hypothesis (LRH) that "model states are a simple sparse sum of these representations," which motivates developing SAEs for interpretability purposes. (Engels et al., 2025; Bricken et al., 2023). However, simple addition alone cannot explain how models understand negation as discussed in the introduction.

### 5.2. Identifying Causal Attention Modules for Negation

In Section 5.1, we know that information about "not" has been moved to the position of "Y." Some attention module must continue to relay information about "not Y" to the output position. We use two patching methods[4] to trace causally important attention modules: (1) Path Patching and (2) Attention Sink Ablation.

**Path Patching**   We apply a modified version of path patching (Wang et al., 2023), in which the attention output is treated as the sender and the output embedding as the receiver. We ignore changes in attention patterns caused by patching following Jafari et al. (2025), focusing on how attention outputs influence the model's final predictions.

Here we formally define the path patching process. For a pair of prompts $(P_+, P_-)$, we first run standard forward passes and record the attention outputs $\mathcal{AO}_\ell(P_+), \mathcal{AO}_\ell(P_-)$ and attention patterns $\mathcal{AP}_\ell(P_+), \mathcal{AP}_\ell(P_-)$ at all layers $\ell$, at the last token position. We then run a path-patched forward pass on the negative prompt $P_-^{pp}$. At a set of target layers[5] $\mathcal{L}_t = \{\ell_1, \ell_2, \ldots, \ell_m\}$ (e.g., layers 12–14), we replace the attention outputs by setting $\mathcal{AO}_\ell(P_-^{pp}) \leftarrow \mathcal{AO}_\ell(P_+)$ for all $\ell \in \mathcal{L}_t$. At all other layers we fix the attention patterns to their original values on the negative prompt, i.e., $\mathcal{AP}_\ell(P_-^{pp}) \equiv \mathcal{AP}_\ell(P_-)$, but we recompute MLP outputs.

Suppose that on the original negative prompt $P_-$, the model correctly prefers the answer $y_-$ over $y_+$, i.e., $\Delta(P_-; y_-, y_+) > 0$. We then record whether the path-patched forward pass satisfies $\Delta(P_-^{pp}; y_+, y_-) > 0$, indicating that the model's preference has flipped from $y_-$ to $y_+$. If some attention modules are causally important, we expect to observe a higher proportion of such flips.

**Attention Sink Ablation**   Attention Sink is a complementary method for ablating attention modules. Instead of *Cumulative Attention Sink*, we sink attention modules in a *window* of layers. Similar to path patching, we apply attention sink only at the last token position and keep $\mathcal{AP}_\ell(P_-)$ fixed at all other layers. We denote by $P_-^{as}$ the model input corresponding to the negative prompt when the attention sink (AS) intervention is applied.

Compared to path patching, sinking attention modules is desirable in two ways. First, it is self-contained to individual prompts and does not require additional information or forward passes. Second, it rules out the tricky discrimination between "loss of causality from $\mathcal{AO}(P_-)$" and "introduction of causality from $\mathcal{AP}(P_+)$." If output switches from $y_-$ to $y_+$ in $P_-^{as}$, only the first case is possible and we have found causally important attention modules.

**Results**   Figure 4 shows the result of our attention sink experiments on Llama-3.1-8B. We find that middle layer attention heads (around layer 14) are causally important for negation understanding, indicated by the sharp accuracy drop. In addition, we see that ablating middle-late layers (around layer 17) improves performance, while ablating much later layers does not interfere with negation understanding; this helps show why the *Cumulative Attention Sink* method from Section 4.2 was successful. Figure 6 and Figure 7 in the Appendix show full results for both Attention Sink and Path Patching on both Llama-3.1-8B and Mistral-7B: these plots match our observations in Figure 4, confirming that mid-layer attention modules are causally

---

[4]For patching purposes, we expand our dataset. See Appendix A.3 for details and discussion.

[5]We patch multiple layers to account for functionally overlapping mechanisms.

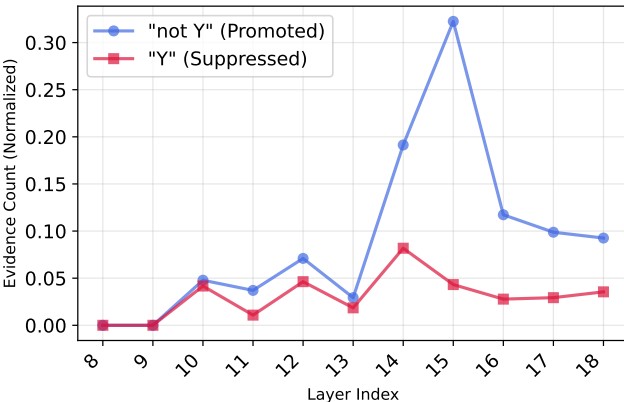

*Figure 5.* Normalized evidence count plotted against attention layer index. The normalized evidence count measures the percentage of samples in the dataset for which evidence is identified at a given layer. Results are on `Llama-3.1-8B`. The blue (red) line indicates the ratio of samples that LogitLens identifies $\bar{Y}$ ($Y$) related tokens as top promoted (demoted) tokens. We observe that evidence count peaks at causally important attention modules.

important for negation processing.

### 5.3. Mid-Layer Attention Moves a Constructed Negated Representation to the Last Token

Recall the two hypotheses about negation: For the phrase "`not Y`," the **construction** hypothesis posits that the negated representation $\bar{Y}$ is explicitly computed and promoted; the **suppression** hypothesis posits that the representation of $Y$ is suppressed. We have shown that mid-layer attention modules are causally important for negation processing. We now study *how* they contribute to the final prediction with the help of LogitLens.

We apply LogitLens on the attention outputs at the last token position of the negative prompt. Tokens with largest logits are deemed as *promoted* tokens. We find that the causally important attention modules promote concepts that are human interpretable and semantically related to "`not Y`." For example, when $Y$ is "`gas`," attention outputs promote "`solid`;" when $Y$ is "`in Asia`," attention outputs promote "`America`"; when $Y$ is "`located near the ocean`," they promote "`inland`." We conclude that mid-layer attention modules encode an explicitly constructed representation $\bar{Y}$ for "`not Y`", which matches the **construction-based** hypothesis. In order to quantitatively test this hypothesis and scale it up, we design an LLM-based annotation pipeline as follows.

**LLM-Based Annotation Pipeline** We first run the model on all $P_-$ and cache $\mathcal{AO}_-$ at the last token position for $\mathcal{L}_{10} \sim \mathcal{L}_{18}$. Then, we use LogitLens to project $\mathcal{AO}_-$ onto the vocabulary. We record the top 10 promoted

tokens for each attention output. After that, we query `openai/gpt-oss-120b` to label whether each token is related to the concept of "`not Y`". Prompt details are provided in Appendix A.4.

**Results** For $> 80\%$ of the examples, the LLM annotator is able to find tokens related to "`not Y`" among the top promoted tokens in at least one layer. We plot the number of examples where at least one token matches "not Y" at each layer in Figure 5. The peak occurs at layer 14, which matches Figure 6.

### 5.4. Mid-Layer Attention Weakly Suppresses

Having demonstrated that mid-layer attention modules promote the negated representation, we apply the same methodology to test if they also directly *suppress* the positive answer. Similar to promotion, tokens with smallest logits are deemed as *suppressed* tokens. For example, when $Y$ is "`Europe`," the attention outputs suppress tokens such as "`Europeans`," "`european`," and "`europe`," which are semantically related to "`Europe`."

**Results** For $> 30\%$ of the examples, the LLM annotator is able to find tokens related to "`Y`". We also plot the number of examples where at least one token matches "`Y`" at each layer in Figure 5. The peak also occurs at layer 14. Suppression is less frequently observed than construction. Combined with Section 5.3, we argue that the model implements both the **construction** and **suppression** hypotheses, with *construction* playing a more central role.

### 5.5. MLPs Promote "`not Y`" Concepts

In Section 5.3, we establish that causally important attention modules construct the negated representation $\bar{Y}$. We now study how the causal signal from attention outputs gets translated to the final negative answer. To this end, we propose a contrastive attribution method to identify important components. We work with `Llama-3.1-8B` in this section because it pairs with a complete set of trained SAEs from He et al. (2024).

**Contrastive Attribution** For a token $t$ in our vocabulary, let $W_U(t)$ denote the row in the unembedding matrix for $t$. For each prompt $P$, we first compute

$$d := W_U(y_-) - W_U(y_+),$$

the direction in representation space that encodes the difference between the negative and positive answers for $P$. We define $\mathcal{C}(x, P)$ as the contribution of model component $x$ to the logit direction $d$ under prompt $P$:

$$\mathcal{C}(x, P) := \langle W_U^\top \mathcal{LN}_{L+1}(x),\ d \rangle.$$

*Table 4.* Representative SAE latents. They are identified by the layer (L) they are applied to and the index number (N). Across tasks, promoted tokens directly instantiate the negated concept, supporting a construction-based implementation of negation.

| Not Y | Mid-Layer Attn. Output | Latent | Top Promoted Tokens |
|---|---|---|---|
| not gases at room temp | solid | L26 / N70467 | `metal`, `silver`, `metallic` |
| not in Asia | America | L22 / N25827 | `England`, `London`, `USA`, `Paris`, `UK` |
| not biodegradable | remaining | L17 / N28594 | `plastic`, `trash`, `litter`, `cleanup`, `plastics` |

Then, we contrast between two runs (e.g. $P_-$ and $P_+$). For any model component $\mathcal{MO}_i$, we compute its *contrastive attribution score* as:

$$\mathcal{C}\left(\mathcal{MO}_i, P_-\right) - \mathcal{C}\left(\mathcal{MO}_i, P_+\right).$$

**Identifying Critical MLPs**  We apply contrastive attribution using two settings: (1) We contrast between $P_-$ and $P_+$, and (2) We contrast between $P_-$ and $P_-^{as}$, where $P_-^{as}$ denotes running the model with attention sinking (§4.2) on $P_-$. Preliminary results show that MLPs tend to score high on the contrastive attribution score. We take the top 10 MLPs from either setting and compute their intersection. This produces a set of critical MLPs after layer 14 for further investigation (roughly $17 \sim 25$).

**Identifying Critical SAE Latents**  After identifying critical MLPs, we apply pre-trained SAEs to help extract critical latent features. For an identified MLP at layer $i$, we apply the corresponding SAE to obtain:

$$\mathcal{MO}_i \approx \sum_{j=1}^{D} \beta_j f_j \tag{3}$$

We then compute the contrastive attribution score for each SAE latent $f_j$ using

$$\mathcal{C}\left(f_j, P_-\right) - \mathcal{C}\left(f_j, P_+\right) \;\; \text{and} \;\; \mathcal{C}\left(f_j, P_-\right) - \mathcal{C}\left(f_j, P_-^{as}\right)$$

This procedure yields a set of critical SAE latents.

**Manual Inspection of Critical Latents**  We manually inspect the top latents identified from the previous step. We use LogitLens to project the latents onto the vocabulary. We record the top tokens promoted and suppressed by each latent as the "explanation" of the latent. Roughly, we manually go over 20 samples and check 50 SAEs per sample. We are able to identify promoting SAEs for 8 samples, with 13 interpretable SAEs in total. At other times, either the SAEs are not interpretable, or the SAE reconstruction error takes full attribution.

First of all, while top promoted tokens are concept-related, top demoted tokens are mostly uninterpretable. This aligns with our findings that construction is stronger than suppression. Secondly, the identified latents directly construct concepts related to "not Y." For example, on the prompt

"`Here is a list of operating systems that are not open source:`", we find latent 31222 at layer 21 promoting "`Win`," "`Windows`," and "`.exe`,". More are given in Table 4.

## 6. Conclusion

In this paper, we mechanistically study how LLMs process negation. First, we demonstrate that *negation* and *shortcut* mechanisms co-exist in LLMs. We hold *shortcut attention heads* accountable for generating incorrect outputs, showcasing that they exhibit biases developed during pre-training. Importantly, we elucidate that **construction** and **suppression** collectively implement negation, where *construction* is the major mechanism. For construction, the model first computes a negated representation $\bar{Y}$ for "not Y" and then promotes output tokens related to it. Concurrently, the representation of "Y" is suppressed. Our work highlights that LLMs are ensembles of competing mechanisms, and that low black-box accuracy can hide the existence of more capable internal mechanisms; thus, fully auditing model capabilities requires thoroughly inspecting model internals, as we do in this paper.

## 7. Limitations

In this work, we focus on the form of negation where an explicit indicator (such as "not") is present. There exists other forms of negation, such as lexical negation (e.g. "unhappy"), adverbial negation (e.g. "seldom") and negation pronouns (e.g. "nobody"). Extending our analysis to other forms of negation is an interesting direction for future work.

## Acknowledgements

This material is based upon work supported by the Defense Advanced Research Projects Agency (DARPA) under Agreement No. HR00112590089. This work was supported in part by the National Science Foundation under Grant No. IIS-2403436. Any opinions, findings, and conclusions or recommendations expressed in this material are those of the author(s) and do not necessarily reflect the views of the National Science Foundation. We also acknowledge support from Coefficient Giving.

## Impact Statement

In this work, we contribute new mechanistic interpretability methods to the research community: Attention Sink Ablation and Contrastive Attribution. We present a complete pipeline that applies various methods to uncover a specific mechanism. Results from our paper deepen the understanding of LLM internals. Our work inspires future research that helps build more robust and reliable LLMs.

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

# A. Experimental Setup

## A.1. Models

We study pre-trained base models from several families, namely Llama3.1 (Grattafiori et al., 2024), Qwen2.5 (Yang et al., 2024), Qwen3 (Yang et al., 2025), Gemma2 (Riviere et al., 2024), Mistral-v0.1 (Jiang et al., 2023) and OLMo2 (Walsh et al., 2025). The models are of size $\sim 7B$. For mechanistic purposes, we study meta-llama/Llama-3.1-8B and mistralai/Mistral-7B-v0.1. We focus on base models, instead of instruction-tuned models, to study mechanisms generally arising from pre-training, avoiding confounds introduced by dependence on task- and application-specific tuning data.

## A.2. Dataset Curation

**Prompt Templates** We use four prompt templates when curating all data entries. We illustrate prompt templating in Table 7. Here the four prompt templates reuse the same factual question: "What is an animal that is not an amphibian?"

**Pipeline** The first step is to manually curate factual question pairs (positive and negative) with annotated gold answers. We use a fixed template for this step (specifically template 1 in Table 7). The number of distinct factual pairs with annotated gold answers is 81.

The second step is to expand the prompts obtained by querying a powerful commercial LLM (in our case gpt-4). Note that every prompt can be categorized uniquely with the corresponding tuple of (X, Y). We take two layers of safeguards to ensure that the expanded prompts are unique. First, we prepend all existing tuples of (X, Y) in the prompt and instruct the LLM to generate new tuples of (X, Y). Second, we use symbolic programs to check that the tuples are unique. Up to this point, we obtain 162 data entries. Finally, we ask the LLM to format each prompt using the four templates. In total, we obtain 648 data entries.

## A.3. Expanded Dataset for Patching

Only for patching, we expand our dataset so that every $P_+$ ($P_-$) has multiple $y_+$ ($y_-$). Once we have the expanded data, we measure the average logits of all negative answers $\overline{y_-}$ and positive answers $\overline{y_+}$ on a prompt. The *surrogate negation accuracy* measures if $\Delta(P_-; \overline{y_-}, \overline{y_+}) > 0$. There are two reasons for this. 1) Using the *surrogate accuracy* on this dataset greatly improves negation accuracy and facilitates locating causally important modules. 2) Using multiple negative answers ensures that the identified modules are important for a general concept, instead of a specific token.

*Table 5.* Prompt subject categories and examples

| Category Type | Examples |
|---|---|
| Living things | animals, plants, trees, fruits, vegetables, insects, mammals |
| Human domains | careers, sports, festivals, languages |
| Technology | apps, operating systems, programming languages, websites |
| Materials & objects | materials, tools, furniture, clothing, instruments |
| Environment | natural disasters, weather, planets |
| Consumables | foods, beverages, drinks |
| Geography | cities, countries |
| Miscellaneous | colors, games, objects |

*Table 6.* Example of one data entry in our dataset. Each data entry contains a pair of prompts: 1) the positive (affirmative) prompt, 2) the negative prompt. Each entry also comes with a pair of gold answers for each prompt.

---

**Positive Prompt:** An animal that is **indeed** an amphibian is a frog.
**Negative Prompt:** An animal that is **not** an amphibian is a dog.

---

| | |
|---|---|
| 1 | Here is a list of animals that are not amphibians: |
| 2 | An animal that is not an amphibian is |
| 3 | Something that is an animal and not an amphibian is |
| 4 | What is an animal that is not an amphibian? It is |

*Table 7.* Illustration of the four prompt templates used for negation sensitivity analysis.

For every stem question template that we have (e.g. "Here is a list of animals that are not (indeed) amphibians"), we ask gpt-4o to generate more positive answers and negative answers.

The positive prompt used is given as the following:

> **System Prompt:**
> You are a helpful assistant that provides concise lists of single-word examples. Each example must be exactly one word. Use lowercase for common nouns, but capitalize proper nouns (countries, cities, brands, programming languages, planets, etc.). Return only comma-separated single words without numbering, bullets, or explanations.
>
> **User Prompt:**
> List exactly 5 ONE-WORD examples of animals that are NOT amphibians. Each example must be a single word in lowercase (except proper nouns like countries, cities, brands, which should be capitalized). Return ONLY a comma-separated list, no explanations or numbering.

The negative prompt used is given as the following:

> **System Prompt:**
> You are a helpful assistant that provides concise lists of single-word examples. Each example must be exactly one word. Use lowercase for common nouns, but capitalize proper nouns (countries, cities, brands, programming languages, planets, etc.). Return only comma-separated single words without numbering, bullets, or explanations.
>
> **User Prompt:**
> List exactly 5 ONE-WORD examples of animals that are amphibians. Each example must be a single word in lowercase (except proper nouns like countries, cities, brands, which should be capitalized). Return ONLY a comma-separated list, no explanations or numbering.

### A.4. Annotation Prompt for Negated Representations

The model is asked to format its output with the following fields: 1) layer id, 2) tokens related to "not Y," and 3) an explanation. We then parse the output and aggregate results across all $P_-$.

```
#### Instruction

You will be given a prompt involving negation
↪   and a list of attention output tokens from
↪   different layers of a language model.

#### Prompt format

The prompt will be of the form **"A that are
↪   not B"**.

Example prompt:

- Here is a list of materials that are not
↪   biodegradable:

In this prompt:
* **A** = "materials"
* **B** = "biodegradable"

#### What to look for
```

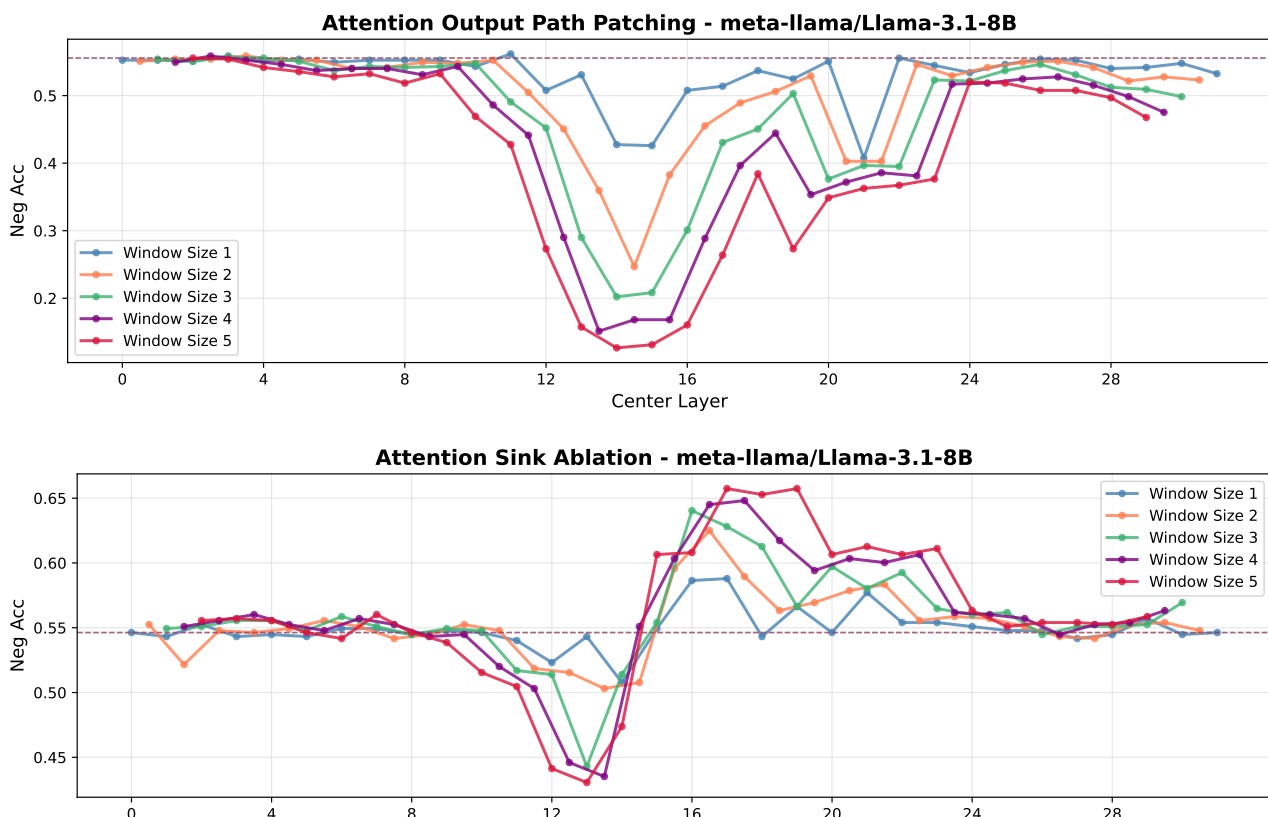

*Figure 6.* Path Patching and Attention Sink Ablation results on Llama-3.1-8B. X axis indicates the center layer that we ablate or patch. Y axis is negation accuracy. Both methods suggest that mid-layer attention modules are causally important for negation processing.

```
Your task is to identify **evidence that the
↪  model represents the semantic concept
↪  "not B"**, rather than merely attending
↪  to unrelated or noisy tokens.

Valid evidence should fall into at least one
↪  of the following categories:

* **Direct antonyms or negations of B**
* **Properties or categories logically
↪  incompatible with B**
* **Concepts that strongly imply not B (via
↪  world knowledge)**

Do **NOT** count:

* Formatting artifacts, punctuation, or
↪  multilingual noise
* Tokens whose connection to not B is
↪  speculative or weak
* General topical tokens unrelated to B

If evidence is ambiguous or uncertain, **do
↪  not include it**.
```

```
#### Example attention outputs
... (Left out for brevity) ...

#### Example reasoning chain

- the negation part in the prompt is "not
↪  biodegradable"
- going through the attention outputs,
↪  `14_attn_out` contains " commercial" and
↪  " Remaining". " commercial" could be
↪  related to unnatural. " Remaining" could
↪  be related to not decomposing.

Side note: Other examples contain easier
↪  cases. For example:
- " solid" relates to "not gas"
- " inland" relates to "not located near the
↪  ocean"
- " American" relates to "not in Asia"

#### Expected output

Output a JSON list. Each entry should include:

* the layer index
```

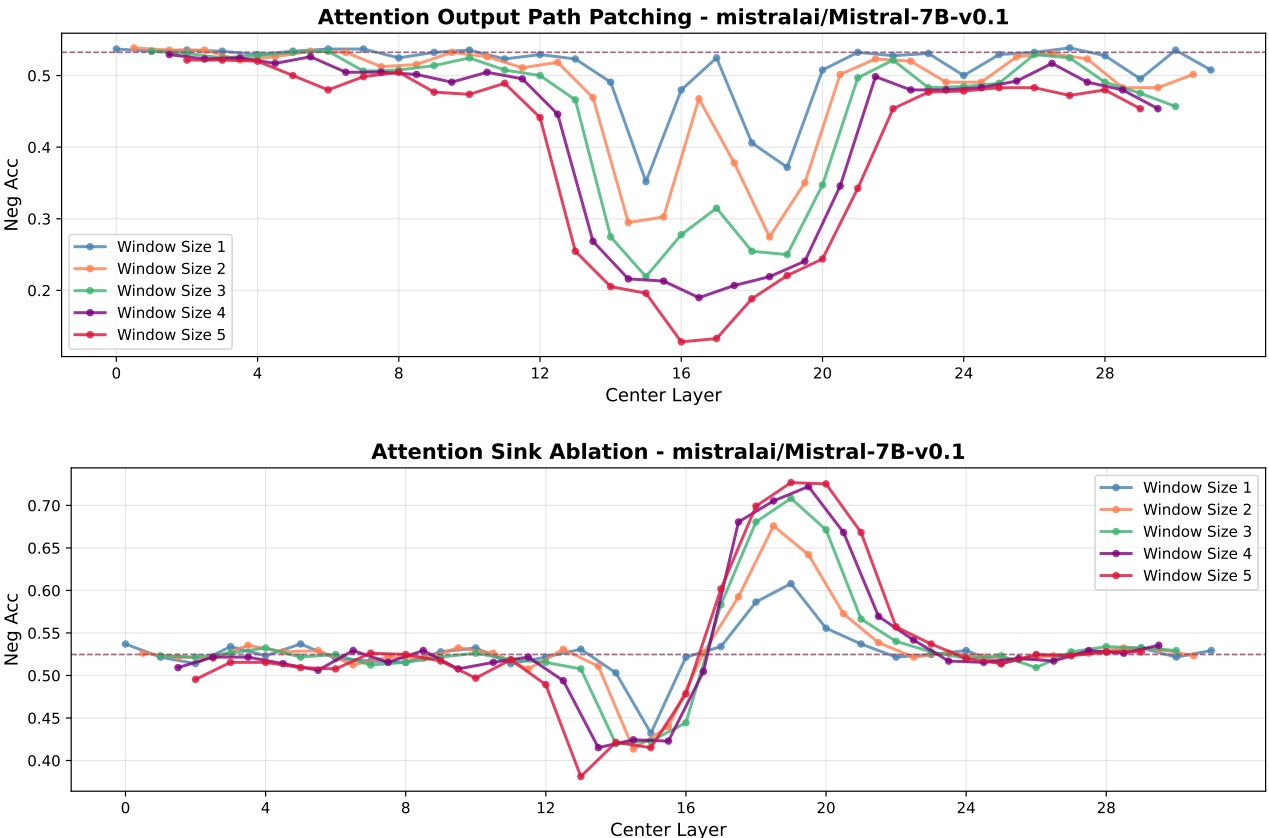

*Figure 7.* Path Patching and Attention Sink Ablation results on Mistral-7B. X axis indicates the center layer that we ablate or patch. Y axis is negation accuracy. Both methods suggest that mid-layer attention modules are causally important for negation processing.

```
* the evidence tokens
* a brief justification explaining why they
↪   indicate "not B"

Example:

```json
[
    {
        "layer": 14,
        "tokens": [" commercial", "
        ↪   Remaining"],
        "justification": "These tokens suggest
        ↪   non-natural, persistent materials,
        ↪   which are incompatible with
        ↪   biodegradability."
    }
]
```

If **no convincing evidence exists**, output:

```json
[]
```
```

## B. Motivation for Cumulative Attention Sink

Here we provide a discussion on the motivation for using *Cumulative Attention Sink* in Section 4.2. The reasons are two-fold. First, we want to match LogitLens conceptually. Second, Section 5.2 suggests that late-layer attention modules are irrelevant for negation processing.

We elaborate more on what we mean by matching LogitLens conceptually. When we apply LogitLens to the hidden state at an intermediate layer, we are effectively zero-ablating all subsequent layers. Following the same spirit, when we apply Attention Sink to ablate attention modules, we want to ablate all subsequent attention modules following an intermediate layer. This is why we apply Attention Sink cumulatively.

We acknowledge that for example a windowed version of Attention Sink could produce better results in recovering negation accuracy. However, we choose the cumulative version for the reasons above and for simplicity. The goal for Section 4.2 is to identify the existence of shortcut attention heads, rather than to exhaust the full potential of attention ablation in recovering negation accuracy.

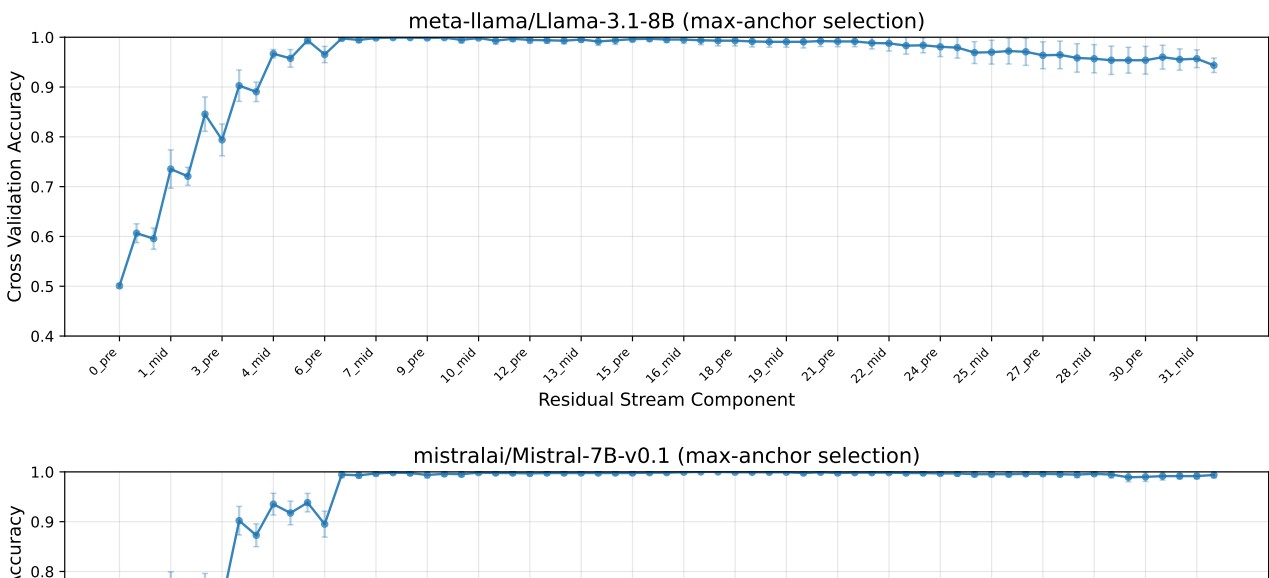

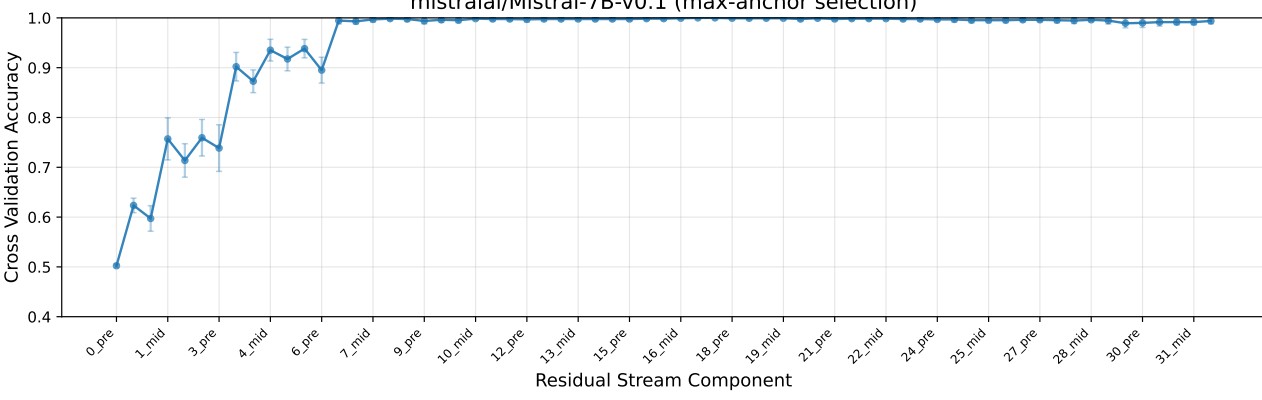

*Figure 8.* Cross-validated LDA model accuracy as a function of position in the residual stream. A higher accuracy indicates that we can decode "not" from the residual stream more reliably. As shown, "not" is moved to "Y" at early to middle layers.

## C. On Attention Sink and Attention Knockout

Geva et al. (2023) propose *Attention Knockout* as a method to ablate attention modules moving information from specific spans of tokens of interest. While implementation-wise our *Attention Sink* is equivalently knocking out all tokens other than the first and current token, our motivation and assumptions are different: we follow Xiao et al. (2024) to come up with an ablation method for attention modules that still makes the model function in a relatively reasonable domain.

## D. Additional Analyses

### D.1. Sanity Check on Negation Sensitivity

Is *sensitivity* defined in Section 4.1 just an artifact of randomness? Is our choice of one canonical $\mathcal{A}_+$ and $\mathcal{A}_-$ reasonable? To rule out these concerns, we conduct the following sanity check.

Let random variable $X$ denote the mean of $\Delta(P_-; y_-, y_+) - \Delta(P_+; y_-, y_+)$, which is the difference of logit differences on $\mathcal{P}_+$ and $\mathcal{P}_-$ on a data entry.

The null hypothesis is that the distribution of $X$ is irrelevant to our selection of $\mathcal{A}_+$ and $\mathcal{A}_-$.

To simulate the distribution under the null hypothesis, we randomly select two *arbitrary* answer tokens for each data entry as a positive and negative answer. We compute the mean $X^*$ over the dataset and repeat this experiment 500 times for each model. The empirical p-value for $X^* > X$ is $< 0.002$ for all models. Therefore, we conclude that models are sensitive to negation and that our answer choices are acceptable; the sensitivity of our chosen exemplar positive and negative answers serves as a proxy for the class of such answers.

### D.2. PCA Visualization of Hidden States

In Figure 11, we plot the PCA results at multiple positions of the residual stream. The position after attention module at layer 11 intuitively best separates two classes.

### D.3. Full Results on Decoding "Not"

In Figure 8, we plot the full accuracy of decoding "not" from the residual stream at different layers. We achieve near

perfect performances at early layers.

### D.4. Full Patching Results

In Figure 6 and Figure 7, we plot the full patching results of path patching and attention sink. Both methods point to the causal importance of middle layer attention modules. Additionally, attention sink reveals that there exists shortcut attention heads and that late layer attention heads are irrelevant.

### D.5. LLM Annotation Results for `Mistral-7B-v0.1`

In Figure 9, we plot LLM annotation results for identifying concepts related to promoting "`not Y`" and suppressing "`Y`" in attention outputs from `Mistral-7B-v0.1`. It shows a similar trend to that of `Llama-3.1-8B`. Additionally, the trend for promoting concepts related to "`not Y`" matches with patching results in Figure 7.

### D.6. Multi-Answer Evaluation

Our main results in Section 4.1 use a single canonical pair of answer tokens $(y_+, y_-)$ per prompt. To verify that our findings are not an artifact of this choice, we re-evaluate all six base models on the expanded dataset described in Appendix A.3, where every prompt is paired with multiple positive and negative answer tokens. For each prompt we average the logits over all positive answers $(\overline{y_+})$ and over all negative answers $(\overline{y_-})$ before computing accuracy and sensitivity.

*Table 8.* Multi-answer evaluation. Negative accuracy, positive accuracy, and sensitivity are computed using averaged logits over multiple candidate answers per prompt. All values are reported as percentages (%) with three significant figures.

| Model | Neg Acc | Pos Acc | Sens. |
|---|---|---|---|
| Llama-3.1 | 54.6 | 95.5 | 98.8 |
| Qwen2.5 | 64.2 | 91.7 | 97.8 |
| Qwen3 | 64.0 | 88.7 | 97.1 |
| Gemma-2 | 55.7 | 96.6 | 98.5 |
| Mistral-v0.1 | 52.8 | 95.8 | 97.8 |
| OLMo-2 | 60.8 | 97.4 | 98.9 |

The multi-answer results in Table 8 closely track the single-answer results in Table 2. Across all six models, positive accuracy remains near-saturated ($\geq 88.7\%$), negative accuracy stays well below positive accuracy, and sensitivity is uniformly high ($> 95\%$). Per-model deviations between the two evaluation protocols are small (typically within $\sim 6$ absolute percentage points on negative accuracy), and the relative ordering of models is preserved. We therefore conclude that the qualitative findings in the main paper – substantial gap between positive and negative accuracy, paired with high sensitivity – are not artifacts of the specific

$(y_+, y_-)$ chosen, but properties of how these models process negation.

### D.7. Best Layers for Attention Sink and LogitLens

Table 3 reports the maximum negative accuracy achieved across all candidate layers when applying *Cumulative Attention Sink* or LogitLens. For completeness, Table 9 lists the best-performing layer index for each method and each model. Layer indices are 1-based and are reported alongside the total number of hidden layers $L$ of each model. Across all six models, the best Attention Sink layer satisfies $> 0.5L$, supporting the claim in Section 4.2 that shortcut modules reside in middle-to-late layers. The best LogitLens layer is similar to or slightly later than the best Attention Sink layer.

*Table 9.* Best layer (1-indexed) for Attention Sink and LogitLens on each model, together with the best negative accuracy (%) achieved at that layer. **Num Layers** is the total number of hidden layers $L$ reported by the model configuration.

| Model | Num Layers | Attn. Sink | | LogitLens | |
|---|---|---|---|---|---|
| | | **Best Acc** | **Best Layer** | **Best Acc** | **Best Layer** |
| Llama-3.1 | 32 | 67.8 | 17 | 53.6 | 30 |
| Qwen2.5 | 28 | 65.4 | 23 | 59.4 | 24 |
| Qwen3 | 36 | 64.2 | 28 | 59.6 | 25 |
| Gemma-2 | 42 | 66.1 | 27 | 59.7 | 27 |
| Mistral-v0.1 | 32 | 65.9 | 18 | 61.6 | 17 |
| OLMo-2 | 32 | 68.7 | 21 | 61.6 | 21 |

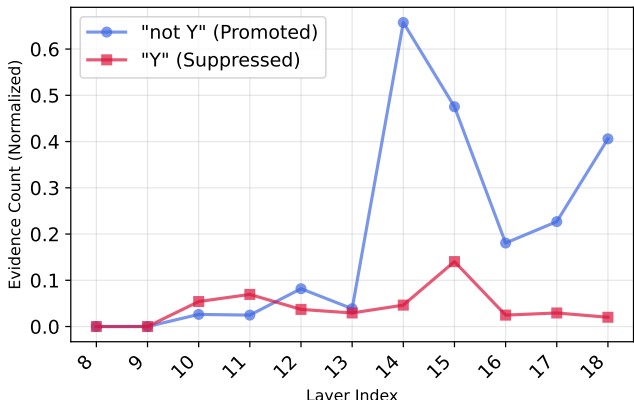

*Figure 9.* Normalized evidence count plotted against attention layer index. Results are on `mistralai/Mistral-7B-v0.1`. The blue (red) line indicates the ratio of samples that LogitLens identifies $\bar{Y}$ ($Y$) related tokens as top promoted (demoted) tokens. We observe that evidence count for "not" follows the same trends as patching results in Figure 7.

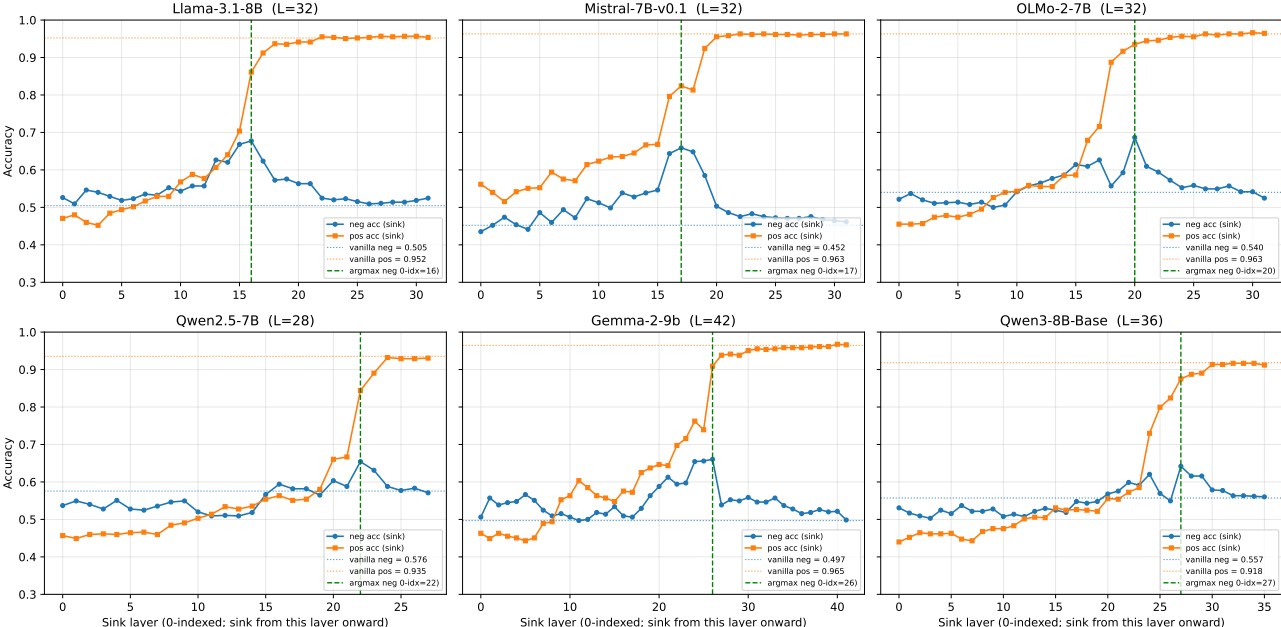

*Figure 10.* Negative and positive accuracies as a function of the sink layer for *Cumulative Attention Sink*. For each model, we sweep the layer from which the sink is applied (0-indexed, $x$-axis) and report both negative accuracy and positive accuracy. Dotted horizontal lines mark the vanilla (no-sink) accuracies, and the dashed green vertical line marks the sink layer that achieves the maximum negative accuracy. Across all six models, attention modules *after* the best negative-accuracy layer affect only positive accuracy, and positive accuracy is already close to its vanilla saturation by that layer.

PCA Overview (2D) — Layers 10-15 — meta-llama/Llama-3.1-8B

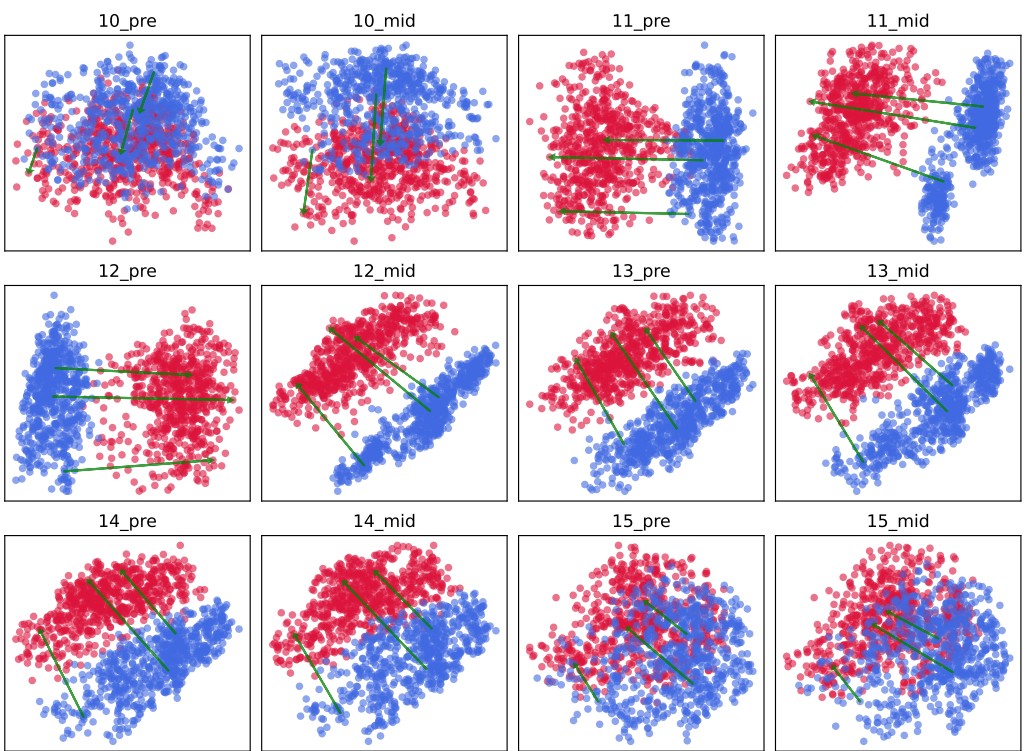

*Figure 11.* Visualization of the PCA space at different model layers. The hidden states of $P_+$ and $P_-$ are colored as blue and red. Arrows indicate the direction from one hidden state of $P_+$ to the corresponding hidden state of $P_-$. It can be seen that positive and negative hidden states are approximately linearly separable by one direction.

