# OpenReview forum: "How Language Models Process Negation"
_ICML.cc/2026/Conference — ICML 2026 regular_

### Official Review · Reviewer_qbYd · 2026-02-25

**Soundness:** 3
**Presentation:** 2
**Significance:** 3
**Originality:** 2
**Overall Recommendation:** 3
**Confidence:** 4

**Summary:**

This paper investigates how multilayer perceptrons handle negation words and why they often fail to recognize them despite their internal sensitivity to "not". The authors argue that negation word computation is achieved through an attention mechanism in the middle layers, which constructs a "not Y" representation, which is then amplified by the LLM layers. The fast attention mechanism in later layers may then override correct reasoning. This analysis combines representation separability, attention sink intervention, and logical attribution to support this hierarchical explanation.

**Compliance With Llm Reviewing Policy:**

Affirmed.

**Final Justification:**

I very appreciate the efforts author made in rebuttal. I think the rebuttal solve most of my questions. However, for my core question, the author didn't directly answer.

For my question 2, I think author can directly use the characteristic of the hidden state which uses PCA to  analysis to prove the credibility of PCA. However, authors use other method to prove PCA is not important. Because of this, I do not understand why use PCA here, if PCA is not suitable and not important, the analysis of Section 5.1 will be mitigated.

Also there are many confused words in the paper, which need to polish.  Because I have already promised the author that I would improve my score, and the author has resolved most of my concerns. So I will raise my score to weak reject.

**Key Questions For Authors:**

- The author mentions the attention sink method in section 4.2. Attention sink now has a relatively accepted definition [2], but the author's method seems to contradict this definition. I think the author needs to consider changing the method name, as it took me a long time to understand its meaning.

- I believe that section 5.1 in the author's description emphasizes the role of the early layers, but its relationship to subsequent layers is not strong. Furthermore, I think there are some fundamental issues in section 5.1, see weakness.

- Overall, I believe the author has provided a very comprehensive analysis of the negation problem, and the conclusions are quite solid. However, I think there are some fundamental errors in the author's argument, but if the author can address my concerns, I am willing to change my rate to "accept".


[1] arXiv:2502.02013

[2] arXiv:2309.17453

**Limitations:**

Yes

**Strengths And Weaknesses:**

## Strengths
- Negation is a long-standing systemic weakness in LLM inference. This paper does not aim to improve benchmarks, but rather attempts to explain "why it goes wrong," suggesting that the problem itself has mechanistic significance.

- This article provides a complete explanation of the negation phenomenon, with sound arguments, rich methodology, and comprehensive results that support the author's viewpoint.

- The experiment was repeated on multiple models, which reduced the risk of "single-model peculiarities".



## Weaknesses
-  I think there are some errors in equation.1, I believe the summation operation should be performed on both AO and MO simultaneously. In equation 1, authors only do this on AO.

-  In sections 4.2 and 5.2, the authors used the attention sink method to demonstrate the importance of intermediate layers for negation and the shortcut properties of subsequent layers. However, it is unclear whether this improvement in rejection is due to a breakdown in model performance. The authors need to conduct more comparative experiments to observe whether positive accuracy remains stable and whether perplexity changes after adopting the attention sink method. I also strongly suspect that this method leads to a breakdown in model performance.


-  In Section 5.1, PCA is used under the condition that the data has a low-rank structure. However, in many models, the hidden states of early layers, such as the first and second layers, do not have the low-rank property. Performing PCA on layers that do not have the low-rank property will result in a significant loss of feature information. I believe the authors should first prove the low-rank property of the hidden state of the Y position.


-  In line 270 of section 5.1, "attention heads move the representation of 'not' to the last token position." However, the author's hidden state is extracted from the residual state, and there is no longer an attention head in the hidden state after attention. Therefore, the concept of attention head moving here is somewhat far-fetched.


-  Similarly, the authors in section 5.3 still use mid-layer attention and window ablation, which fails to prove that the attention head is related to negation.


-  The authors' findings that the impact of negation is more significant in the intermediate layer are not new. [1] and other articles have proposed that models with stronger semantic understanding capabilities in the intermediate layer have been proposed.

---

> ### Author Rebuttal · Authors · 2026-03-31
>
> Thank you for recognizing our effort to present a comprehensive view of negation mechanisms and for your constructive feedback. Below, we respond to the identified weaknesses.
>
> # KQ1: Concerns regarding using the term "Attention Sink"
> We cite [Xiao et al.](https://openreview.net/forum?id=NG7sS51zVF) for our so called "Attention Sink" method. They term the tokens that absorb large attention weights as "attention sink tokens" (typically the first token). In our paper, we restrict the attention to only attend to itself and the first token. We find our method conforming to its definition.
>
> If this does not address the concern, we are happy to change our method name to "Restricted Attention Masking", or would greatly appreciate any other suggestions.
>
> # KQ2: Concerns regarding early layers in Section 5.1
> We have new experiment results that validate that early layers in Section 5.1 are causally important and worth studying. We acknowledge that we do not study how "not" + "Y" gets further computed to $\bar{Y}$, and leave this as exciting future work.
>
> The setting is as follows. For all tokens of "Y" as query tokens, we mask out the "not" token in the key tokens for attention heads in a very large window size 25 for 32 layer models. The last token can still attend to "not" and aggregate all information.
>
> If we mask out the "not" token in early layers, there is a significant drop in accuracy. However, if we mask out all layers 8-32, there is almost no drop in accuracy. This suggests that the "not" token is bound to "Y" in early layers. These results also converge with our current results and Figure 8, which shows that we achieve near perfect linear decoding accuracy around layer 7.
>
> Here are the drops in negation accuracy for llama and mistral when applying attention masking at layers of window size of 25 starting from layer 1-8.
> ```
> Llama: 0.414, 0.423, 0.392, 0.336, 0.225, 0.157, 0.108, 0.025
> Mistral: 0.327, 0.324, 0.290, 0.259, 0.161, 0.120, 0.096, 0.034
> ```
>
> ## Other concerns in weaknesses
> 1. Thank you for pointing out. We will change to $ h_{L + 1} = E + \sum\limits_{i = 1}^{L} (\mathcal{AO}_{i} + \mathcal{MO}_{i})$.
> 2. We provide a control experiment to show that Attention Sink does not break the model.
>
> We record the best layer to apply Cumulative Attention Sink. We then apply at the same layer to the positive prompts. As shown in the table below, the sinked models still achieve $\sim 90\%$ positive accuracy. The drop in positive accuracy is negligible compared to the improvement in negation accuracy.
>
> |Model|#Layers|Attn Sink Layer|NegAcc(Vanilla)|NegAcc(Sink)|ΔNegAcc|PosAcc(Vanilla)|PosAcc(Sink)|ΔPosAcc|
> |---|---|---|---|---|---|---|---|---|
> |Llama-3.1-8B|32|17|0.522|0.728|+0.207|0.951|0.920|-0.031|
> |Qwen2.5-7B|28|23|0.639|0.688|+0.049|0.941|0.901|-0.040|
> |Qwen3-8B-Base|36|28|0.633|0.710|+0.077|0.914|0.880|-0.034|
> |Gemma-2-9B|42|27|0.546|0.682|+0.136|0.985|0.957|-0.028|
> |Mistral-7B-v0.1|32|19|0.438|0.661|+0.222|0.963|0.935|-0.028|
> |OLMo-2-7B|32|20|0.648|0.765|+0.117|0.957|0.923|-0.034|
>
> We do not claim that Attention Sink should be adopted during model generation. Rather, we argue with Attention Sink experiments that we identify the existence of shortcut attention heads.
>
> Here is a qualitative result that shows Attention Sink does not break the model.
>
> ```
> Prompt: Here is a list of foods that are indeed desserts:
> Top tokens:
> POS: [' ice', ' cake', ' chocolate', ' cakes', ' cookies', ' apple', ' pie', ' Ice', ' brown', ' chees']
> NEG: [' ice', ' a', ' apples', ' bread', ' cake', ' pizza', ' ', ' fruits', ' breakfast', ' chocolate']
> NEG_SINK: [' a', ' bread', ' food', ' ', ' pizza', ' breakfast', ' cereal', ' eggs', ' vegetables', ' meat']
> ```
>
> Applying attention sink at layer 17 removes " ice", " cake", " chocolate", which are likely desserts.
>
> 3. We first perform PCA to gain insights on our dataset. Unsupervised PCA results reveal the possibility that information about the token "not" could be linearly decodable. We then verify this hypothesis with further LDA experiments. Therefore, we did not impose low rank structure hypotheses on the data. Instead, we just found that there exists such a structure that positive and negative prompts can be separated by a single direction.
> 4. In Section 5.1, we conclude so because attention is the only module that can move information between tokens. We find that we can decode information about "not" from the residual stream. We are open to better interpretation of this result.
> 5. In Section 5.3, we follow [Yu et al.](https://aclanthology.org/2024.emnlp-main.193/) to trace causally important attention modules by tracking accuracy drops.
> 6. We are happy to discuss this in related works. [1] mainly analyze the richness or semantic content of representations at different layers, rather than explaining the coexistence of correct intermediate computation and incorrect final prediction.

---

> > ### Author Rebuttal · Reviewer_qbYd · 2026-04-01
> >
> > Thanks authors response, however, I think my questions are not fully resolved. Here is some reasons
> >
> > - I appreciate authors' efforts to prove attention sink doesn't break the model. However, in my question, I want author show the perplexity. But, in this turn, I won't see this concerns as my main concerns, because current result has already prove the points.
> >
> > - I think the authors may have misunderstood my question. I am not asking about the assumptions required for PCA; rather, I am questioning whether applying PCA here is justified. If the data are not low-rank, PCA may discard a substantial amount of information, which would be a serious issue for the downstream analysis. Although the later analyses are consistent with the authors’ claims, I believe that, as the first step of the analysis, the authors need to provide sufficient motivation for why this method is appropriate. **This is always my core concern**.
> >
> > - The authors seem to conflate head level information transfer with residual stream level observations. The current evidence only shows that a negation related signal is present in the residual state at the Y position; it does not directly establish the stronger head-level claim that attention heads “move” the representation of “not” there.
> >
> > Overall, I would like the authors to address my questions directly. My third weakness concerns the motivation behind the design of the analysis, while my fourth concerns the rigor of the writing in the paper. If the authors insist that the residual hidden states contain attention heads, I would regard that as a factual misunderstanding.

---

> > > ### Author Response · Authors · 2026-04-05
> > >
> > > Thank you for your additional feedback for helping us make our paper clearer. We try to address your concerns below.
> > > # Concerns regarding applying PCA
> > > We understand your concern that PCA may discard important information. We are willing to incorporate your advice and make Section 5.1 more rigorous with two steps:
> > > - **Directly training a linear probe achieves accuracies with a similar trend**:
> > >
> > > We provide additional linear probing results that eliminates low-rank hypotheseses (if any) introduced by PCA. We apply almost the same pipeline to train linear probes as in Section 5.1, but without applying PCA. We perform the same 10-fold cross validation. We train an LDA model for each layer on the training set and evaluate on the validation set. We provide figures in the same style as Figure 8. Here are the [result for Llama-3.1-8B](https://anonymous.4open.science/r/ICML26_negation_rebuttal_qbYd-67C7/full_lda_acc_original_vs_attn_ablated_meta-llama--Llama-3.1-8B.png) and [result for Mistral-7B-v0.1](https://anonymous.4open.science/r/ICML26_negation_rebuttal_qbYd-67C7/full_lda_acc_original_vs_attn_ablated_mistralai--Mistral-7B-v0.1.png).
> > >
> > > - **Qualifying our claim on PCA visualization**:
> > >
> > > We will properly qualify our claim with respect to PCA visualization. We acknowledge that we only apply PCA to suggest visual intuitions. We follow [Rimsky et al.](https://aclanthology.org/2024.acl-long.828/), who also apply PCA to "assess the degree of linear separability of the internal representation".
> > >
> > > # Concerns regarding attention-level arguments
> > > We are again willing to incorporate your advice on attention-level arguments and make our paper more rigorous.
> > >
> > > - **Focusing on objective results**:
> > >
> > > We will focus on honestly laying out the objective experiment results. First, with our addition linear probing results, we argue that it is linearly decodable at early layers of the residual stream whether the hidden state is from the positive or negative prompt.
> > >
> > > Next, we argue that attention heads are causally affecting linear separability with our new experiment results. Our general pipeline is almost the same as our previous linear decoding analysis. First, we can obtain linear probes trained on the positive and negative prompts from this pipeline.
> > >
> > > We then take the negative prompt and mask out negation indicator token "not" in all attention heads. We then use the trained linear probes to classify whether the hidden state is from the positive prompt or the intervened negative prompt. The accuracy drops to approximately random chance of 50%. This accuracy is labeled as "attn ablated" in our [result for Llama-3.1-8B](https://anonymous.4open.science/r/ICML26_negation_rebuttal_qbYd-67C7/full_lda_acc_original_vs_attn_ablated_meta-llama--Llama-3.1-8B.png) and [result for Mistral-7B-v0.1](https://anonymous.4open.science/r/ICML26_negation_rebuttal_qbYd-67C7/full_lda_acc_original_vs_attn_ablated_mistralai--Mistral-7B-v0.1.png).
> > >
> > > From our experiments, we find that 1) it is linearly decodable at early layers of the residual stream whether the hidden state is from the positive or negative prompt, and 2) attention heads are causally affecting linear separability. We will ensure a rigorous description of just the objective experiment results in our revised paper.

---

### Official Review · Reviewer_Z4g5 · 2026-03-04

**Soundness:** 3
**Presentation:** 2
**Significance:** 4
**Originality:** 3
**Overall Recommendation:** 4
**Confidence:** 4

**Summary:**

This paper focuses on investigating the mechanism of how negation is processed in modern LLMs. Their findings include:

- Models can handle negation internally, but incorrect answers are caused by late-layer attention heads that somehow force the model to generate non-negated answers.
- Models likely handle negation by directly constructing negated concepts like "not gas → solid" in most cases, achieved by first adding the representation of "not" to the negation target "gas" and then promoting this negated representation, accompanied by a less dominant mechanism of suppressing the negated concept like "gas."

**Compliance With Llm Reviewing Policy:**

Affirmed.

**Key Questions For Authors:**

These are the main reasons I lean toward weak rejection, as they might affect the validity of Section 4. If these concerns are addressed, I am willing to raise my score toward accept.

1. The results in Section 4 rely on single-token logits (L136), but I cannot find a complete list of target answers or verification that every answer is tokenized into a single token across all models' tokenizers, since different tokenizers are likely to tokenize differently.
2. Do you compare the logits at the first token position after the prompt in all settings? I am not convinced this is sufficient to prove that models cannot handle negation. You need to rule out the case where models intend to generate outputs like **"An animal that is not an amphibian is a 1. reptile. 2. fish. 3. insect. 4. mammal."** which I obtained directly from Llama-3.1-8B using your prompts. In this case, the model handles negation correctly, and it is entirely understandable if the model assigns a higher logit to the wrong answer at the first token position. A simple way to rule out this possibility is to report the logit comparison as a function of token index, with the x-axis being the token index and the y-axis being the positive/negative accuracy.

**Limitations:**

This paper also needs to explicitly discuss the limitations of its results, for example, the limited dataset size (648 entries), the stacked assumptions underlying the SAE analysis in Section 5.5, etc.

**Strengths And Weaknesses:**

### Strengths
- This paper provides a well-structured and novel mechanistic account of how models handle negation in Section 5. The finding that models first move the representation of "not" to the negated target and then promote this mixed representation is well supported by solid experimental results.
- The Attention Sink method is a clean and simple ablation technique.
- This paper utilizes multiple methods (Attention Sink, path patching, LogitLens) to identify causally important layers, and the consistent results across methods strengthen the reliability of the mechanistic account.


### Weaknesses
- The presentation of this paper needs revision. Currently, the introduction sets up Suppression vs. Construction as the main research question, but the paper then spends Section 4 on a completely different question: whether models are able to handle negation at all, before returning to Suppression vs. Construction in Section 5. This makes Sections 4 and 5 feel like two separate mini-studies. I suggest rewriting the introduction to match the actual flow.
- The claim in L45 - 46 that prior work suggests two competing hypotheses needs citations to support it.
- Projecting SAE latents back via LogitLens involves a stack of assumptions: the linearity assumption of LogitLens, the fixed layer norm assumption, and SAE reconstruction fidelity. I understand this is common practice in the MI community, but this limitation should be explicitly stated somewhere. Additionally, the number of SAE latents inspected is too small (20 samples) and the results are weak (only 8 interpretable latents).
- The naming of "shortcut" comes from "they pick up spurious features (e.g., co-occurrence)" in lines 80 - 81, but the paper does not show that these heads pick up spurious features. Instead, it only shows that suppressing certain attention heads ("sinking" them) helps the model predict the correct answer. This alone is not evidence of a shortcut mechanism, and I suggest changing the terminology.
- Section 4 may have a significant oversight; see my key questions below.

---

> ### Author Rebuttal · Authors · 2026-03-31
>
> Thank you for recognizing our mechanistic account of negation and for your constructive feedback. Below, we respond to the identified weaknesses.
>
> # KQ1 Concerns regarding single token logits evaluation
> We handle this in our evaluation script carefully. For all data entries, we compute the first diverging token between the positive and negative answer. Shared prefix becomes part of the prompt. We additionally check programmatically that >90% of our dataset has positive and negative answers both tokenized into a single token for all other models (one outlier is 51% for Mistral 7B). Overall, we carefully ensure the soundness of our experiments.
>
> # KQ2 Concerns regarding evaluation at the first token position
> First, we don't argue that models cannot handle negation. We report in Table 1 that these LLMs have ~30% lower accuracy on negative prompts compared with positive prompts. Our research question here is why LLMs correctly process negation (high sensitivity), but it is overshadowed by another mechanism.
>
> Second, we kindly argue that the first generated token logit can already reflect the model’s ability. In your case, "An animal that is not an amphibian is a 1. reptile. 2. fish. 3. insect. 4. Mammal.", the model will have high logits for all possible candidates at the “reptile” position. Then, referring to [Yan et al.](https://aclanthology.org/2025.emnlp-main.815/), the model will suppress “reptile” in the following generations and complete the generation. It is suggested in the review that the model would assign higher logits to wrongs answers at the first token position as compared to at following positions, but this mechanism is outside the scope of our current paper.
>
> We provide a qualitative result to help resolve your concerns. For the prompt "Here is a list of foods that are not desserts:", the top tokens from llama at the first token position are:
> ```
> Positive Prompt: [' ice', ' cake', ' chocolate', ' cakes', ' cookies', ' apple', ' pie', ' Ice', ' brown', ' chees']
> Negative Prompt: [' ice', ' a', ' apples', ' bread', ' cake', ' pizza', ' ', ' fruits', ' breakfast', ' chocolate']
> Negative Prompt with Attention Sink: [' a', ' bread', ' food', ' ', ' pizza', ' breakfast', ' cereal', ' eggs', ' vegetables', ' meat']
> ```
>
> At the first token position of the negative prompt, the model promotes " ice", " cake", " chocolate", which are likely desserts. However, applying Attention Sink at layer 17 removes them, resulting in much cleaner top tokens.
>
> # Other concerns in the weaknesses section
> 1. The introduction is written with careful contemplation. First of all, the main theme of our paper is how LLMs handle negation. This is why we set out clearly in the introduction explaining our core hypotheses. Then, we deliberately include contents from Section 4 to honestly present the nuances of interpretability research. This is because experimentally, we must first verify that there exists some negation mechanism in the models to be revealed before applying causal tracing and interpretabilities methods.
> 2. The following paragraph, L58 - L68, immediately resolves your concerns
> 3. We are happy to state the full assumptions behind interpreting SAE latents. We wanted to correct a misunderstanding here. We manually checked 20 samples x 50 SAEs / sample = 100 SAEs. We found Evidence for 8 / 20 samples, with 13 interpretable SAEs. Interpretability is not a guaranteed blessing. We made our best effort to find evidence that supports our claim.
> 4. We provide more results to help resolve your concerns on shortcut mechanisms. We apply the same Attention Sink to the positive prompts. As shown, we achieve consistent improvements in negation accuracy, but consistent drops in positive accuracy, quantitatively supporting that the shortcut attention heads are promoting positive answers on both positive and negative prompts.
>
> |Model|#Layers|Attn Sink Layer|NegAcc(Vanilla)|NegAcc(Sink)|ΔNegAcc|PosAcc(Vanilla)|PosAcc(Sink)|ΔPosAcc|
> |---|---|---|---|---|---|---|---|---|
> |Llama-3.1-8B|32|17|0.522|0.728|+0.207|0.951|0.920|-0.031|
> |Qwen2.5-7B|28|23|0.639|0.688|+0.049|0.941|0.901|-0.040|
> |Qwen3-8B-Base|36|28|0.633|0.710|+0.077|0.914|0.880|-0.034|
> |Gemma-2-9B|42|27|0.546|0.682|+0.136|0.985|0.957|-0.028|
> |Mistral-7B-v0.1|32|19|0.438|0.661|+0.222|0.963|0.935|-0.028|
> |OLMo-2-7B|32|20|0.648|0.765|+0.117|0.957|0.923|-0.034|
>
> Here is a qualitative result that shows shortcut attention behaviors.
> ```
> Example 1:
> Prompt: Here is a list of animals that are not domesticated:
>
> 28_attn_out Top Tokens:
> POS: [' dogs', ' dog', 'dog', ' Dogs', 'dogs', ' Dog', 'Dog', ' here', ' canine', ' horses']
> NEG: [' here', 'here', 'Here', ' Here', ' HERE', ' animals', ' dogs', ' horses', ' aquí', ' certain']
> # comment: dogs and horses appear in both positive and negative prompt outputs.
> ```

---

> > ### Author Rebuttal · Reviewer_Z4g5 · 2026-04-01
> >
> > All my key concerns have been addressed, assuming the clarification on the multi-token evaluation procedure will be integrated into the final version. I now lean toward accept, as this paper makes an insightful contribution to understanding how models handle negation. I have raised my score accordingly.

---

> > > ### Author Response · Authors · 2026-04-05
> > >
> > > Thank you again for your service, and thank you for raising your score.
> > >
> > > We will make sure to incorporate clarification on the multi-token evaluation procedure into the final version.

---

### Official Review · Reviewer_MDk5 · 2026-03-04

**Soundness:** 2
**Presentation:** 2
**Significance:** 3
**Originality:** 3
**Overall Recommendation:** 5
**Confidence:** 4

**Summary:**

This paper investigates how language models (LMs) process negation. The authors find that while LMs struggle with negation prompts, their logits indicate they are still sensitive to negation, suggesting there are competing mechanisms at play. They show attention sinking of later attention layers can improve performance on negation sentences because they incorrectly promote related positive concepts. Next, they characterize the suppression of related concepts and promotion of negated concepts and how they both contribute to negation predictions.

**Compliance With Llm Reviewing Policy:**

Affirmed.

**Final Justification:**

The authors' initial rebuttal and subsequent reply have appropriately answered my questions and helped clarify the design choices they have made. I believe this extra context and explanations will strengthen the paper and make it easier to read, and trust they will make it into the final version of the paper.
Based on this, my only unresolved concern is that there is a key missing piece, which is understanding how "not Y" is converted into $\bar{Y}$. I agree with the authors that it is probably too much for this paper to also investigate, but seems like it would help us understand negation more generally. Based on the updates and clarification from the authors, I recommend accepting this paper.

**Key Questions For Authors:**

Many of my questions are in the weaknesses question, but here are two others:

1. How confident are you that the PCA direction separating your data described in Section 5 represents "not"? Have you tried using it to steer a word Y's representation to be more like "not Y" or more like "Y"?

2. In the Appendix you mention you use an expanded dataset for patching. If it works better for locating what you're looking for, why not just use the expanded dataset in general?

**Limitations:**

More discussion of the limitations of the analysis would be good. I've noted some examples in the Weaknesses section above (e.g., limited scope/interpretation of negation, etc.)

**Strengths And Weaknesses:**

**Strengths:**

- The topic of study is interesting. Negation continues to be a challenge for modern language models (as well as VLMs + Text-to-Image Models that depend on LMs), and progress towards addressing this has the potential to be impactful.
- The sensitivity result in Section 4.1 are nice. They show that models are more capable of understanding negation than what raw accuracy captures.
- An attempt is made to show the generality of the effects across several models (Qwen, Llama, Mistral).

___

**Weaknesses:**

I have several concerns with the paper, but in general the experiment design and methodological decisions felt ad-hoc, several experimental details are missing, and some conclusions are made without sufficient evidence to support them. There are also some lines of work related to negation that were not mentioned that might help contextualize these findings better. I've tried to describe these below:

- Experimental details are missing/unclear:
  - While the results presented in Table 2 are nice, they need more context to be understood. For example, for which layers is the attention sink being applied? Lines 252-253 mentions "For all models, we record the layer at which applying Cumulative Attention Sink achieves the best performance". Do you use the same layer for LogitLens as well? While it appears you've run this experiment for many different values, I can't find it presented anywhere in the paper or appendix.
  - The dotted line in Figure 4 is not labeled. Is it cumulative attention sink accuracy, or something else? It's also unclear what the y-axis of Figure 4 is accuracy of. Is it negation accuracy over your 600+ prompt dataset?
  - Some details related to dataset construction were unclear, specifically how answer tokens were chosen. The data example they provide: “An animal that is not an amphibian is a dog” - why is dog chosen as a “gold” label? There are many other potentially correct answers like "mammal", "cow", etc. Is this what you're trying to measure against in Appendix B.1? It wasn't quite clear.
  - Line 282: “last token position corresponding to Y” - This phrase is a bit confusing. Can you elaborate what this means? Last token position usually means end of the prompt, but are you referring to the last token of the (potentially multi-token) phrase Y (it seems like this is potentially the case based on examples in lines 340-343, right column)?
  - Line 750, right (section B.1) - As far as I can tell $A_{+}$ or $A_{-}$ are not defined anywhere prior to this. From context, it seems like these refer to positive and negative answers chosen from some set of plausible tokens, but this was not clearly explained in the dataset curation phase (A.2). Do you use the responses from template 1 as a pool for candidate answers to a prompt, or something else? Why is this different from $y_+$, $y_-$ notation used earlier?

- Lack of sufficient evidence to support some conclusions:
  - The form of negation studied in the paper is narrow compared to how broadly negation can be expressed in language. Lines 126-128 mention "We study a controlled family of prompts of the form 'X that is not Y is Z'." The dataset is small (~650 entries), and there are other forms of negation that this does not really capture like morphological negation (e.g. unhappy, or atypical) or negation of verbs (it seems like all cases you look at are common nouns?). So it's unclear what this tells us about negation broadly beyond this simple dataset.
  - The cumulative attention sink experiment is not very targeted, and because of this it's unclear how much we can glean from it. Lines 255-256 say "The best layer is consistently > 0.5L, suggesting that shortcut modules reside in middle-to-late layers", but as far as I can tell there is no data presented about how layers compare to one another for the cumulative attention sink experiments (4.2).
  - The PCA results suggest the positive and negative prompts are separable (as verified by the LDA classifier), but it's not clear to me based on the experiments whether it’s a “not” direction. Using this direction to steer the representation of Y to be either more like “Y” or “not Y” might help support/strengthen this claim. For example, Marks et al. similarly use negation datasets and find a PCA direction that separates samples in the way you describe here, but they claim the direction represents a "truth" direction (I am aware their setup is different and they're not studying negation directly).
  - Lines 442-443: "we contribute new mechanistic interpretability methods to the research community". Can you elaborate on what the specific methodological contribution of this paper is?
- “We show that the bias introduced by shortcut attention modules can be traced back to pre-training” (Lines 190-192). This seems to be vacuously true? (i.e. everything the model learns can be “traced back to training” in some way). However, it doesn't seem like the results are tied back to the shortcut attention modules at all, it appears to only measure accuracy broadly. Given this, the results/claims in section 4.3 were not particularly convincing to me as to why we should connect them to shortcut attention modules.

- Ad-hoc Experiment/Methodological Choices. Sometimes the experiment or method choices felt somewhat arbitrary. Providing more context to help the reader understand why a certain method was chosen or how a hypothesis was formed would strengthen the writing.
  - The paper states "We defer discussion on the motivation for [cumulative attention sinks] to Section 5.2." (Lines 215-216, Right Column). This statement felt detrimental to the flow of the paper, and a simple justification would suffice. (e.g. "we use cumulative attention sinking to measure whether attention is the primary contributor to shortcut behavior and compare to LogitLens, another early-exit decoding strategy that skips all layers instead of just late-layer attention computation.") However, after having read 5.2, the cumulative attention sink decision is not very satisfying compared to the more targeted windowed attention sink. Especially because it seems like windowed attention sinking works better? (i.e. Figure 4 shows you get close to 95% around layer 17)
  - A comparison to between attention sinking and attention knockouts, which has been used in prior work (e.g. Geva et al.) would help strengthen the choice to use a new method.
  - Lines 304-306: “This suggests that early attention layers move “not” to the position of “Y,” laying a foundation for further composition.” This seems to be a claim about information flow, and seems like it could be verified with more standard techniques like patching experiments or targeted attention knockouts. Any reason why you chose to use PCA analysis instead to establish a claim about information flow?
  -  Line 202-204, (right column) says “We suspect that late-layer attention modules exhibit shortcut behavior in those layers of the model.” Can you provide some more background here? What’s the intuition behind how you came to suspect this?

- Related work not discussed:
  - Negation has been studied quite a bit in the context of NLI. In particular, it is known that several NLI datasets (such as SNLI) contain artifacts that prior generations of LMs could rely on to predict entailment of negated statements. See [Gururangan et al.](https://aclanthology.org/N18-2017/) , or [Poliak et al.](https://aclanthology.org/S18-2023/) for more background.
  - Marks et al.'s geometry of truth paper also studies negation, but with the purpose of studying true/false statements. Their probing work feels relevant to your PCA analysis.
  - Halawi et al. is a prior work that shows ablating heads can improve performance because they promoting something harmful for the task of interest, mirroring your findings of "shortcut mechanism" for negation.
  - Elhelo et al. is a prior work that has previously shown that some attention heads directly promote the *antonyms* of words. Perhaps these are connected to the middle-layer or the shortcut heads you find in this study.
___
- Gururangan et al. Annotation Artifacts in Natural Language Inference Data. (https://aclanthology.org/N18-2017/)
- Poliak et al. Hypothesis Only Baselines in Natural Language Inference. (https://aclanthology.org/S18-2023/)
- Geva et al. Dissecting Recall of Factual Associations in Auto-Regressive Language Models. (https://aclanthology.org/2023.emnlp-main.751/)
- Elhelo et al. Inferring Functionality of Attention Heads from Their Parameters. (https://aclanthology.org/2025.acl-long.866/)
- Halawi, et al. Overthinking the Truth: Understanding how Language Models Process False Demonstrations (https://openreview.net/forum?id=Tigr1kMDZy)
- Marks et al. The Geometry of Truth: Emergent Linear Structure in Large Language Model Representations of True/False Datasets (https://openreview.net/forum?id=aajyHYjjsk)

___
Here are some typos I caught while reading:
- Line 311: “conforms” -> confirms?
- Line 323: “continual to” -> continue to?
- Line 616: “intruct-tuned” -> instruct-tuned

---

> ### Author Rebuttal · Authors · 2026-03-31
>
> Thank you for finding our topic interesting and for your constructive feedback. Below, we respond to the identified weaknesses.
>
> # KQ1: Concerns regarding Section 5.1 PCA results
> We first establish that the early-layer behavior in Section 5.1 is causally important with our new experiments. Then, we argue that training a linear probe to detect information flow is the standard approach as in [Li et al.](https://aclanthology.org/2021.acl-long.143/) Our PCA analysis serves two purposes: 1) to provide visual insights, and 2) to perform unsupervised feature selection and avoid overfitting. Finally, success in steering is not guaranteed because of binding issues. There should be mechanisms binding "not" to "Y" instead of "X". Just adding the embedding of "not" to "Y" could be an ideal simplification. We leave steering/control as exciting future work.
>
> The setting for our new experiment is as follows. For all tokens of "Y" as query tokens, we mask out the "not" token in the key tokens for attention heads in a very large window size (e.g. 25 for a 32 layer model). However, the last token can still attend to "not" and aggregate all information.
>
> What we find is that if we mask out the "not" token in early layers, there is a significant drop in accuracy. However, if we mask out all layers 8-32, there is almost no drop in accuracy. This suggests that the "not" token is bound to "Y" in early layers. These results also converge with our current results in Section 5.1 and Figure 8, which shows that we achieve perfect linear decoding accuracy around layer 7.
>
> Here are the drops in negation accuracy for llama and mistral when applying attention masking at layers of window size of 25 starting from layer 1-8.
> ```
> Llama: 0.414,0.423,0.392,0.336,0.225,0.157,0.108,0.025
> Mistral: 0.327,0.324,0.290,0.259,0.161,0.120,0.096,0.034
> ```
>
> # KQ2: Evaluation with Expanded Dataset
> We used the expanded dataset with multiple answers for patching because we want to capture the mechanism regarding "not Y" to $\bar{Y}$, instead of generating actual instances of "not Y". We did not use it for evaluation in Section 4 because around 80% of the answers are generated by AI. That said, we found that results on the expanded dataset match the results in Section 4. Columns "Single T1" and "Multi T1" are comparisons of Table 1 results on single/multiple answer datasets.
>
> |Model|#L|AS|LL|SingleT1|MultiT1|
> |------------|--|----|-----|--------------|--------------|
> |Llama-3.1-8B|32|17|29|52.2/95.1/97.5|56.5/96.3/99.4|
> |Qwen2.5-7B|28|23|23|63.9/93.8/97.2|68.2/93.8/98.2|
> |Qwen3-8B|36|28|34|63.3/91.4/94.8|66.4/89.2/96.9|
> |Gemma-2-9B|42|27|37|54.6/98.8/98.5|60.2/98.2/99.7|
> |Mistral-7B|32|19|16|43.8/96.3/96.3|56.5/97.8/98.2|
> |OLMo-2-7B|32|20|20|64.8/95.7/99.1|67.6/98.5/99.4|
>
> # Other weaknesses
> ## Experimental details
> We will incorporate the following clarifications in the revised paper, either in the appendix or in the main text.
> 1. In the table above, we present the layers at which we apply Attention Sink and LogitLens in Section 4.2.
> 2. The y-axis in Figure 4 is the neg acc over our 600+ prompts. The dotted line is the vanilla neg acc over our 600+ prompts without patching.
> 3. The question-answer pairs are written by the authors based on commonsense world knowledge. We acknowledge that there could be multiple answers, and we have shown that muli-answer evaluation closely matches the single-answer evaluation.
> 4. Yes, the last token of the (potentially multi-token) phrase Y.
> 5. We will fix and use $y_{+,-}$
> ## Sufficient evidence
> 1. We need very controlled prompts for mechanistic studies. For example, [Wang et al.](https://openreview.net/forum?id=NpsVSN6o4ul) uses 15 prompts to generate synthetic data. We highlight that our prompts are not solvable by summing up the fact.
> 2. We have the figures for this to include in appendix. There is a clear peak around layer 17 for Llama.
> 3. See KQ1
> 4. Attention Sink and Contrastive Attribution in our specific settings
>
> ## Tracing Pretraining
> We use the term "shortcut mechanism" in 4.3. We show that there is a bias. This is different from random, which is about 50\% accuracy.
>
> ## Adhoc choices
> 1. We try to match Attention Sink and LogitLens. LL zero ablates accumulatively. If we perform windowed AS, we should do windowed zero ablation. Still, we only argue that shortcut exists and currently it is sufficient.
> 2. We don't argue that AS is better than attention knockouts.
> 3. We address this in KQ1.
> 4. We manually inspect top promoted tokens. For "Animals that are indeed domesticated:"
> ```
> [28_attn_out]
> POS: [' dogs',..., ' horses']
> NEG: [' here',' animals', ' dogs', ' horses', ' aquí', ' certain']
> ```
> dogs and horses appear in both positive and negative prompt outputs.
> ## Related work
> Thank you for the suggestions. We will incorporate them in the revised paper.

---

> > ### Author Rebuttal · Reviewer_MDk5 · 2026-04-02
> >
> > I thank the authors for their responses, I think the additional details will make the paper more clear to read.
> >
> > While I appreciate the explanation regarding PCA to identify the "not direction", I do not think it fully addresses my concern.
> > It seems clear that from your experiments that the "not" token is moved to the "Y" token's hidden state in early layers.
> > However, just having information flow does not tell us *how* the model actually processes the negation of "Y", and there is a significant speculation towards the end of section 5.1 that is not well-supported with evidence. I would recommend updating this paragraph (Lines 308-317). I don't think there's clear evidence explaining how "Y" is converted into "not Y". While middle layers seem to output a vector corresponding to "not Y" it's not clear how/if this composition happens earlier (in early layers), or is done by the middle layers themselves. This is a key missing explanation that would strengthen the claim made in the title you've explained "how LLMs process negation".  I do not think this is a trivial computation to understand. But without this, the contributions of the paper are somewhat moderate; shortcut heads have been seen before in other settings (e.g Halawi et al.), and the choice of methods is still somewhat unconventional with minimal justification.
> > ___
> >
> > In your rebuttal you mention: "We don't argue that AS is better than attention knockouts."
> >
> > This was not my concern. I simply mentioned that comparing to prior methods or at least justifying the use of a new method (attention sinks) for tracing information flow would strengthen the paper, and I still agree with this.
> >
> > ___
> > Overall, I still have concerns related to the choice of methods, some of the conclusions made based on not enough evidence, and design of the analysis as a whole.
> > I would be willing to update my score to weak accept if the authors can adequately motivate their design choices, and address these issues more explicitly.

---

> > > ### Author Response · Authors · 2026-04-05
> > >
> > > Thank you for your additional feedback that helps to make our paper clearer.
> > > # Comparison with prior methods for tracing information flow
> > > We will include results from three additional methods for tracing information flow. The additional methods are:
> > > - **Attention Knockout**: From [Geva et al.](https://aclanthology.org/2023.emnlp-main.751/) For our specific setting, we mask out tokens in the span of "not Y" at the last token.
> > > - **Zero Ablation**: From [Wang et al.](https://openreview.net/forum?id=NpsVSN6o4ul) We set attention outputs to zero.
> > > - **Mean Ablation**: From Wang et al. For every attention layer and each entry, we compute the mean representation of all other entries at the same layer and replace the original attention output.
> > >
> > > We find converging evidence on the importance of mid-layer attention for negation. We provide figures in the same style as Figure 6 and 7 in the [anonymous repo](https://anonymous.4open.science/r/ICML26_negation_rebuttal-081C) for llama and mistral.
> > >
> > > # Specific motivations for Attention Sink
> > > - **A large portion of vanilla attention weights is given to the first and the current token**
> > >
> > > Here are the stats for attention weights after softmax at the last token position on our dataset. We measure what % of attention weights is given to the first and the current token.
> > > |Model|1st+current(%)|
> > > |---|---|
> > > |Llama-3.1-8B|0.7935|
> > > |Qwen2.5-7B|0.6793|
> > > |Qwen3-8B-Base|0.7024|
> > > |Gemma-2-9B|0.6792|
> > > |Mistral-7B-v0.1|0.7784|
> > >
> > > We find that keeping only the first and current token would preserve most attention weights.
> > > - **Prior work suggests that sink tokens are important**
> > >
> > > [Xiao et al.](https://openreview.net/forum?id=NG7sS51zVF) term tokens with large attention weights as *sink tokens* (typically the first token). We cite Xiao et al. as our motivation for Attention Sink, who show that it is important to keep attending to sink tokens. Moreover, [Gu et al.](https://openreview.net/forum?id=78Nn4QJTEN) show that sink tokens emerge out of optimization due to the softmax function. We take inspiration from prior work, respecting the fact that attending to the first token could be the "default" behavior of attention heads.
> > >
> > > - **Cumulative Attention Sink provides similar intuitions as LogitLens**
> > >
> > > Cumulative Attention Sink work very similar to LogitLens, but it preserves the causal chain between the current residual stream and later MLP layers. In our experiments, it provided good intuitions.
> > >
> > > For example, when we run Llama-3.1-8B with the prompt "Here is a list of foods that are not desserts:", if we directly take the residual stream representation at the last token position at layer 17 and apply LogitLens, we get the following top tokens:
> > > ```
> > > ['aget', 'ophobic', 'nodoc', 'ombat', 'BAT', 'ushima', ' Porno', ' flakes', ' none', 'HECK']
> > > ```
> > > Yet, if we apply Cumulative Attention Sink at layer 17, we get the following top tokens:
> > > ```
> > > [' a', ' bread', ' food', ' ', ' pizza', ' breakfast', ' cereal', ' eggs', ' vegetables', ' meat']
> > > ```
> > > Notice that Cumulative Attention Sink provides much more interpretable results, suggesting that the model already has enough information to answer the question at layer 17.
> > >
> > >
> > > ## Discussion with respect to `attention knockout`
> > > We will discuss thoroughly and directly compare `attention sink` and `attention knockout`. We acknowledge that implementation-wise, `attention sink` is equivalent to `attention knockout` that knocks out all tokens but the first and the current token.
> > >
> > > While Geva et al. introduced attention knockout, they do not emphasize the importance of sink tokens. We provide quantitative results and qualitative intuitions to motivate our design of Attention Sink.
> > >
> > > # Addressing concerns regarding "not Y" --> $\bar{Y}$
> > > Thank you for raising this concern. We will acknowledge this explicitly in the paper that we have not fully investigated the mechanism of how "not Y" --> $\bar{Y}$.
> > >
> > > We would like to kindly argue that this mechanism is a quite generic research topic on its own, most similar to the line of work on *factual recall*. [Meng et al.](https://proceedings.neurips.cc/paper_files/paper/2022/hash/6f1d43d5a82a37e89b0665b33bf3a182-Abstract-Conference.html) locate mid-layer MLPs as important. Geva et al. find early layer MLPs enrich the subject and that attention heads extract the answer.
> > >
> > > Answering how the mapping from "not Y" to $\bar{Y}$ mechanistically is a step further in this direction. We will include discussions on this topic as future work. Possibly, this would be doable via causal tracing the exact components responsible and studying sparse surrogate models (such as SAEs) to interpret the exact mechanism implemented.
> > >
> > > We are happy that our results on shortcut heads match previous findings. Again, the scope of this paper is to present an honest view of how LLMs process negation.

---

### Official Review · Reviewer_azrD · 2026-03-12

**Soundness:** 4
**Presentation:** 3
**Significance:** 4
**Originality:** 4
**Overall Recommendation:** 5
**Confidence:** 4

**Summary:**

This paper uses a range of mechanistic interpretability techniques to study how LLMs process negation; e.g., given the sentence “An animal that is not an amphibian is a ___”, how do LLMs decide what to put in the blank? The authors first show that, although LLMs often make behavioral errors when filling in such blanks, a more nuanced look at their logits shows that they are indeed sensitive to negation - albeit imperfectly. The authors then identify “shortcut attention heads” that mask the underlying ability of models by drawing them away from correct solutions that they could otherwise compute. The authors then compare two hypotheses - construction vs. suppression - for how LLMs might process negation by producing a representation for “not X”. Under construction, the model first encodes “not X” and then uses it to construct concepts that are not X. Under suppression, the model suppresses representations relating to X. Using an attention-sink-based method as well as some SAE-based methods, the authors conclude that models do both construction and suppression, with construction dominating.

**Compliance With Llm Reviewing Policy:**

Affirmed.

**Final Justification:**

The rebuttal has reinforced my prior assessment of the paper as strong, so I am retaining my high score. Since the soundness, originality, significance, and clarity were all high in my assessment, the weighting of them is not important, as any weighting would yield a high score.

**Key Questions For Authors:**

I don’t have key questions  - I found the paper very clear!

**Limitations:**

yes

**Strengths And Weaknesses:**

Strengths:
S1 (significance, originality): The choice of negation as a domain is an excellent one that the authors motivate well. Further, it is also well-motivated to study it using interpretability, contrasting with prior primarily behavioral work.

S2 (soundness): I really like the nuanced behavioral experiments that the authors start with; the usage of more sensitive metrics adds nuance to the existing picture from the literature which has often concluded that LLMs simply cannot process negation. I also appreciated the thoughtfulness in analyzing the data - e.g., the appendix analysis ruling out the possibility that the results could just be driven by randomness.

S3 (soundness): The main, interpretability-based experiments are also very strong. They use a large number of well-motivated methods; the extensive nature of the experiments provides converging evidence for the paper’s main claims.

S4 (soundness, significance): The paper is framed around hypotheses (construction vs. suppression), which is useful for giving it a coherent high-level framing. This framing is a useful contribution in its own right as it might help inspire related future work.

S5 (presentation): The paper is clearly written - or at least as clearly as can be expected given the large number of methods that needed to be explained.

Weaknesses:
W1 (soundness): The hypothesis developed in the paper does not address some aspects of what it means to process negation. Specifically, it does not identify how the model recognizes which token is being negated. However, this isn’t a big concern - it’s beyond the scope of the paper’s (already extensive) set of experiments.

W2 (soundness): Relatedly to W1, the paper also does not address how models construct the “not X” sets. Similarly to W1, though, this is not a big concern as it goes beyond the scope of the paper.

---

> ### Author Rebuttal · Authors · 2026-03-31
>
> Thank you for your appreciation of our work and for your constructive feedback. Below, we respond to the identified weaknesses.
>
> ## W1 Identify how the model recognizes which token is being negated
>
> We agree that an important open question is how the model determines which token is being negated, i.e., why “not” applies to Y rather than X in examples where multiple candidate tokens are present. This corresponds to the parsing + binding problem: how the model identifies the correct grammatical target of negation.
>
> As you note, our paper is not focused on resolving this target-selection question. Instead, we study where negation information is represented and propagated, and provide additional causal evidence about when “not” must interact with the negated token. Our results suggest that this interaction occurs before the final-token aggregation, indicating that the model does not simply defer the decision to the last token.
>
> However, we do not explicitly characterize the mechanism by which the model selects Y instead of X as the negation target (e.g., via specific attention patterns or syntactic cues). Understanding how the model distinguishes between multiple candidate tokens and binds “not” to the correct one is an important direction for future work. We will clarify this limitation in the revision.
>
> ## W2 Mechanism of constructing the "not X" set
>
> We agree that understanding how the model constructs the negated representation $\bar{X}$ is an important question. To investigate how this representation is constructed, we conduct a new experiment that provides evidence on this process.
>
> **New experiment (attention masking).**
> For all tokens of `X` as query tokens, we mask the “not” token in the key positions for attention heads within a large layer window (e.g., 25 layers in a 32-layer model). The last token is still allowed to attend to “not”, so global aggregation remains possible.
>
> **Results.**
> Masking “not” in early layers leads to a significant drop in negation accuracy, whereas masking it only in later layers (e.g., layers 8–32) causes almost no drop.
>
> The drops in negation accuracy for masking windows starting at layers 1–8 are:
>
> - LLaMA: 0.414, 0.423, 0.392, 0.336, 0.225, 0.157, 0.108, 0.025
> - Mistral: 0.327, 0.324, 0.290, 0.259, 0.161, 0.120, 0.096, 0.034
>
> These results suggest that the interaction between “not” and `X` is causally important in early layers and cannot be deferred entirely to the final-token aggregation. If the model only combined “not” and `X` at the last token, masking early-layer access from `X` to “not” should not significantly affect performance, which is inconsistent with our observations.
>
> This evidence indicates that the construction of a negated representation begins in early layers, consistent with Section 5.1 and Figure 8, where linear decodability emerges around layer 7.
>
> Based on these findings, we hypothesize that the mapping from “not X” to $\bar{X}$ may arise from feature enrichment in early MLP layers, which have been shown to store and transform semantic features ([Geva et al., 2023](https://aclanthology.org/2023.emnlp-main.751/)). We will include this hypothesis and further analysis in the revised version, and view identifying the precise components responsible for constructing $\bar{X}$ (e.g., via causal tracing or SAE-based decomposition) as an important direction for future work.

---

> > ### Author Rebuttal · Reviewer_azrD · 2026-04-03
> >
> > Because the weaknesses that I identified were minor, I am happy with them being addressed via adding discussion in the paper, as the authors have suggested. I am retaining my (high) score.

---

> > > ### Author Response · Authors · 2026-04-05
> > >
> > > Thank you again for your service, and thank you for your positive evaluation of our work.

---

### Decision · Program_Chairs · 2026-04-30

**Decision:**

Accept (regular)

**Comment:**

The paper studies how LLMs process negation and fail at it when they do. They identify later-layer attention shortcuts as a culprit; they also identify two distinct mechanistic routes by which models produce negations.

Negation can be a pain point with language models and the reviewers appreciate this as an important problem to study. Reviewers also found value in how the paper is framed/motivated in the form of hypotheses (`azrD`,`Z4g5`) and in studying negation via interpretability rather than just behaviorally (`azrD`).  The experiments are nuanced and strong (`azRD`) and comprehensive in testing across multiple models (`Mdk5`, `qbYd`) via multiple interpretability methods (`Z4g5`, `azrD`, `qbYd`) and the attention sink analysis is clean and simple (`Z4g5`). Many technical concerns (e.g., some missing experimental details, connection between the claims and the experiments) were resolved during the rebuttal.

A few concerns seem to remain:
- the broad claims made in the paper are derived from the dataset of limited size (600 examples) whose diversity is not enough to compensate for this (due to the templated nature of the data); this was noted by a couple of reviewers, and when brought up in internal discussions, this was flagged as a pending concern.
- the paper unfortunately seems to have missed multiple important pieces of related work. We strongly recommend that the authors discuss them well and contextualize the paper more carefully.
- A prominent concern that remains is of `qbYd`. One is regarding the legitimacy of the PCA results; in internal discussions, other reviewers say that the probing and attention knockout experiments alleviate this concern. Another is regarding some wordings/presentation of results, which other reviewers say can be fixed if the authors were careful in their presentation. I hope that the authors take a careful look at these points in future versions of the manuscript.

Overall, this is an interesting lens into studying negation in language models; barring the concerns (at least the latter two of which can be addressed), we recommend accepting the paper.